# `Occult`: Optimizing Collaborative Communications across Experts for Accelerated Parallel MoE Training and Inference

Shuqing Luo [1]   Pingzhi Li [1]   Jie Peng [1]   Yang (Katie) Zhao [2]   Yu (Kevin) Cao [2]   Yu Cheng [3]   Tianlong Chen [1]

## Abstract

Mixture-of-experts (MoE) architectures could achieve impressive computational efficiency with expert parallelism, which relies heavily on all-to-all communication across devices. Unfortunately, such communication overhead typically constitutes a significant portion of the total runtime, hampering the scalability of distributed training and inference for modern MoE models (consuming over $40\%$ runtime in large-scale training). In this paper, we first define *collaborative communication* to illustrate this intrinsic limitation, and then propose system- and algorithm-level innovations to reduce communication costs. Specifically, given a pair of experts co-activated by one token, we call them as *collaborated*, which comprises 2 cases as *intra-* and *inter-collaboration*, depending on whether they are kept on the same device. Our pilot investigations reveal that augmenting the proportion of intra-collaboration can accelerate expert parallel at scale. It motivates us to strategically o̲ptimize c̲ollaborative c̲ommunication for acce̲le̲rat̲ed MoE training and inference, dubbed **`Occult`**. Our designs are capable of either delivering exact results with reduced communication cost, or controllably minimizing the cost with collaboration pruning, materialized by modified fine-tuning. Comprehensive experiments on various MoE-LLMs demonstrate that `Occult` can be faster than popular state-of-the-art inference or training frameworks (more than $1.5\times$ speed up across multiple tasks and models) with comparable or superior quality compared to the standard fine-tuning. Code is available at https://github.com/UNITES-Lab/Occult.

---

[1]The University of North Carolina at Chapel Hill [2]University of Minnesota Twin Cities [3]The Chinese University of Hong Kong. Correspondence to: Tianlong Chen <tianlong@cs.unc.edu>.

*Proceedings of the $42^{nd}$ International Conference on Machine Learning*, Vancouver, Canada. PMLR 267, 2025. Copyright 2025 by the author(s).

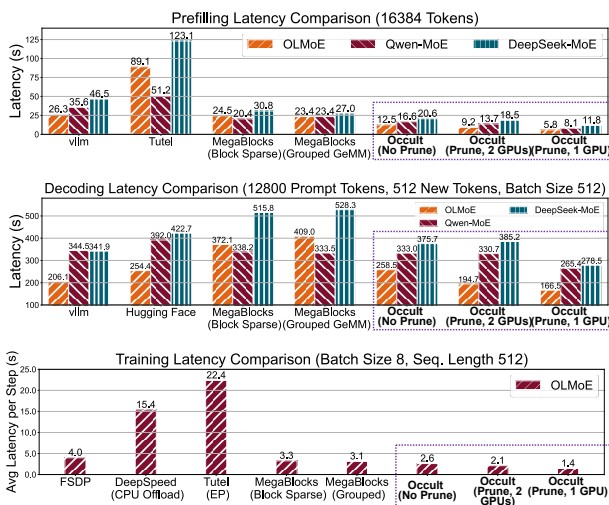

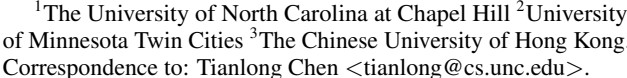

*Figure 1.* **Latency Comparison with Multiple Models & Tasks.** `Occult` can accelerate training & inference for modern MoE-LLMs on communication-intensive tasks.

## 1. Introduction

Transformer-based (Vaswani, 2017) large language models (LLMs) (Achiam et al., 2023; Touvron et al., 2023) have demonstrated tremendous success in a broad spectrum of downstream tasks, and scaling up model parameters is the *de facto* approach to train more powerful LLMs (Jiang et al., 2024; Liu et al., 2024b). Mixture-of-experts (MoE) empowers LLMs with efficient parameter scaling via sparsity. It replaces the standard feed-forward network with a group of experts, and each token only activates a subset, which significantly improves computation efficiency.

However, the giant scale of MoE parameters and the limited memory of a single GPU make it necessary to shard an MoE layer across multiple GPUs for parallelized training and inference. Among the common techniques, expert parallelism is a scalable and efficient approach (Lepikhin et al., 2021; Fedus et al., 2022), which distributes experts on multiple GPUs and dispatches tokens to the corresponding devices via all-to-all communication. As illustrated in Figure 2(a), we can outline a typical forward pass as *Dispatch → All-to-All → Computing → All-to-All → Combine*.

Previous work has demonstrated that all-to-all communication is the main bottleneck (Li et al., 2023a) for both training and inference, especially under heavy workloads (Hwang et al., 2023), where it can count for over $40\%$ runtime. The reason is that communication across GPUs is much slower than computation on a single one. GPUs on the same machine are typically connected via PCIe or NVLink, where the former provides low bandwidth, and the latter is higher but only available on some cutting-edge devices. This makes communication efficiency a pivotal subject for scalable MoE deployment. For modern LLMs, heavy workload is a common teaser, usually emerging in the following tasks:

★ Training: Large batch can better utilize hardware resources, accelerating both pre-training and fine-tuning.

★ Long Sequence Prompting (Prefilling Stage of Inference): A single forward pass can batch massive tokens.

★ Concurrent Generation Requests (Decoding Stage of Inference): Requests from different users are batched together, and decoding is implemented step by step.

The above motivates us to explore how to optimize communication across devices to achieve lower latency and higher throughput. Previous MoE libraries (Hwang et al., 2023; Gale et al., 2023) typically make $k$ replicas for each token in expert parallelism, and dispatch them to different devices, constituting the content of all-to-all communication. In fact, not all content is necessary: if a subset of activated experts for one token is kept on the same device, then only one replica would be required for this device. In the ideal case, if the activated experts are all kept on the same device, then the all-to-all cost would turn out to be almost negligible.

To implement this efficient communication, we develop **a novel sparse matrix multiplication kernel tailored for efficient all-to-all**. Note that if only reusing previous libraries, the streamlined token replicas for efficient communication cannot directly serve as the input/output of expert computing, otherwise extra memory footprint would be required. Therefore, we build a new sparse matrix multiplication kernel for both forward and backward passes in Section 3.1, aiming at reducing unnecessary memory allocation or access. Based on this, expert placement turns out to be critical for communication efficiency: an ideal placement should make the routed experts for each token fall into as few devices as possible.

To this end, we propose to reformulate all-to-all in MoE workflow as *Collaborative Communication*. Given expert $\mathtt{E}_i$ and $\mathtt{E}_j$ co-activated by a token $x$, we denote them as collaborated. Furthermore, expert collaboration can be divided into two categories: (1) Intra Collaboration: if $\mathtt{E}_i$ and $\mathtt{E}_j$ are kept on the same device, and (2) Inter Collaboration: if $\mathtt{E}_i$ and $\mathtt{E}_j$ are kept on different devices.

This collaborative perspective inspires us with innovative solutions to advance MoE communication efficiency:

*Table 1.* **Approximated Linear Correlation between $C_\mathcal{T}$ and runtime.** $2^{14}$ prompt tokens at the prefilling stage are examined.

| OLMoE | MegaBlocks | Occult Trivial | Occult Rescheduled | Occult Prune, 2 GPUs | Occult Prune, 1 GPUs |
|---|---|---|---|---|---|
| $C_\mathcal{T}$ | 8 | 4 | 4 | 2 | 1 |
| $\mathbb{E}(C_\mathcal{T})$ | 8 | 3.68 | 3.02 | 1.98 | 1 |
| Intra / Inter | 0.33/0.67 | 0.33/0.67 | 0.46/0.54 | 0.60/0.40 | 1/0 |
| Latency (s) | 24.50 | 15.93 | 12.53 | 9.22 | 5.82 |

★ For intra collaboration, only one replica of $x$ is required in the *Dispatch* stage. For *Combine* stage the output of $\mathtt{E}_i$ and $\mathtt{E}_j$ can be pre-aggregated locally.

★ For inter-collaboration, typically the token should be duplicated twice. But if it can be transformed into intra-collaboration, only one replica would be required.

Therefore, the communication cost for expert parallelism can be optimized via maximizing intra-collaboration and minimizing inter-collaboration. To formulate it, we first propose a simple and practical criterion to measure the communication budget, *i.e.*, the average number of replicas for each token, denoted as $C_\mathcal{T}$. In Table 1, we present the strong linear correlation between $C_\mathcal{T}$ and runtime. To push the efficiency of expert parallel to the limit, Occult first **reschedules expert placement leveraging profiled collaboration**. We utilize the statistics of intra- and inter-collaboration from a profiling dataset to derive a rescheduled placement, which delivers around $20\%$ reduction on $C_\mathcal{T}$ and runtime, as shown in Table 1. Furthermore, we propose **collaboration pruning for controllable and minimized $C_\mathcal{T}$**, which pushes communication efficiency to the limit controllably.

As an algorithm-system co-design scheme, Occult can accelerate MoE training and inference with expert parallelism on communication-intensive tasks. Our contributions are:

★ **Novel Perspective for Efficient Expert Parallel**. We propose to view the all-to-all communication in expert parallelized MoE workflow from a novel collaborative perspective, and propose an algorithm-system co-design scheme dubbed Occult to accelerate both training and inference for MoE-LLMs.

★ **Higher Quality than Top-$k$ Routing**. We evaluate the performance of MoE-LLMs tuned with collaboration pruning on extensive benchmarks, where Occult can achieve comparable or better performance than top-$k$ routing with improved computational efficiency.

★ **Faster Training and Inference**. Occult achieves wall-clock speedup for training and inference with MoE-LLMs on communication-intensive tasks. 3 frontier models and 3 tasks, including prefilling, decoding, and training, are examined, with an outlined comparison in Figure 1. Occult consistently surpasses other libraries and popular frameworks to varying degrees.

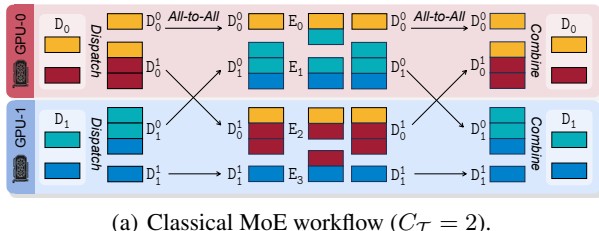

(a) Classical MoE workflow ($C_{\mathcal{T}} = 2$).

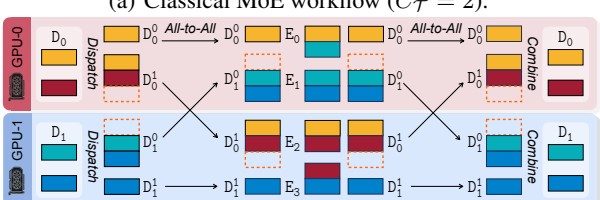

(b) `Occult` workflow w/o collaborative pruning ($C_{\mathcal{T}} = 1.5$).

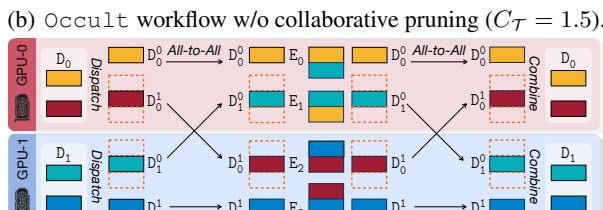

(c) `Occult` workflow w. collaborative pruning ($C_{\mathcal{T}} = 1$).

*Figure 2.* **MoE workflows with 3 all-to-all communication strategies.** We take 2 devices ($D_0$ and $D_1$) for expert parallel, and $D_i^j$ denotes tokens from device $i$ dispatched to device $j$.

## 2. Preliminary

### 2.1. Mixture-of-Experts and Expert Collaboration

Given an input token embedding $\boldsymbol{x}$, the output of an MoE layer is the weighted sum of outputs from the $N_e$ experts $\{E_0, \ldots, E_{N_e-1}\}$:

$$\text{MoE}(\boldsymbol{x}) = \sum_{i=0}^{N_e-1} \mathcal{R}(\boldsymbol{x})_i \cdot E_i(\boldsymbol{x}), \qquad (1)$$

where $\mathcal{R}(\boldsymbol{x})_i$ is the output of router network $\mathcal{R}(\cdot)$ for the $i$-th expert. For each token, the MoE layer aggregates the output of $k$ experts based on the top-$k$ highest scores obtained from the *Softmax* value of a gating function $g(\cdot)$, which is usually a single linear projection layer:

$$\mathcal{R}(\boldsymbol{x}) = \text{Top-K}(Softmax(g(\boldsymbol{x})), k),$$

$$\text{Top-K}(\boldsymbol{v}, k) = \begin{cases} \boldsymbol{v}, & \text{if } \boldsymbol{v} \text{ is in the top } k, \\ 0, & \text{otherwise.} \end{cases} \qquad (2)$$

To quantify the interactions among experts, we construct a graph $\mathcal{C}$ based on the expert co-activation patterns within a token batch $\mathcal{B}$. Each edge represents the co-activation times between 2 co-activated experts:

$$\mathcal{C}_{i,j} = \sum_{\boldsymbol{x} \in \mathcal{B}} \mathbb{1}\{\mathcal{R}(\boldsymbol{x})_i \neq 0 \wedge \mathcal{R}(\boldsymbol{x})_j \neq 0\}. \qquad (3)$$

We further normalize all edge values in $\mathcal{C}$ by dividing by the maximum edge value, yielding a matrix $\mathcal{P} \in \mathbb{R}^{N_e \times N_e}$ with values in the interval $[0, 1]$:

$$\mathcal{P}_{i,j} = \mathcal{C}_{i,j} \Big/ \max_{0 \leq i,j \leq N_e-1} \mathcal{C}_{i,j}. \qquad (4)$$

### 2.2. Measuring All-to-All Communication Complexity

We propose to measure the all-to-all communication complexity with $C_{\mathcal{T}}$, motivated by 2 aspects:

1. In the *Dispatch* stage, typically $k$ replicas are prepared for each token, serving as the all-to-all content. An individual replica may be reserved locally or transmitted to another GPU, based on the routing choices.

2. Exactly measuring the communication volume leads to varied evaluations on different datasets. $C_{\mathcal{T}}$ can tackle this issue, since it is the supremum of inter-device all-to-all communication volume, making it task-agnostic.

As shown in Table 1, $C_{\mathcal{T}}$ exhibits a strong linear correlation with wall-clock runtime, since it is highly related to communication overhead and memory footprint. Based on this empirical relationship, we formulate the boundaries of the all-to-all communication budget: Consider a distributed system with $N_e$ experts and $N_d$ devices implementing top-$k$ routing, where $N_e$ is divisible by $N_d$. We define $\underline{C_{\mathcal{T}}}$ and $\overline{C_{\mathcal{T}}}$ as the lower and upper bounds of $C_{\mathcal{T}}$ respectively, where:

$$1 \leq \lceil \frac{k \cdot N_d}{N_e} \rceil \leq \underline{C_{\mathcal{T}}} \leq \overline{C_{\mathcal{T}}} \leq \min\{k, N_d\}.[1] \qquad (5)$$

Therefore the theoretically optimal budget for all-to-all communication can be formulated as $\underline{C_{\mathcal{T}}} = \lceil \frac{k \cdot N_d}{N_e} \rceil$.

### 2.3. Optimizing All-to-All Communication Complexity

In Figure 2, we progressively optimize $C_{\mathcal{T}}$ to improve communication efficiency of expert parallelism. Figure 2(a) shows the conventional expert parallelism. A key observation is that not all tokens require $k$ replications: the red and green tokens necessitate only a single copy, since both replicas are transmitted to the same device. Figure 2(b) depicts how leveraging this insight reduces the communication cost $C_{\mathcal{T}}$ from 2 to 1.5. Nonetheless, the yellow and blue tokens still incur duplicate transmissions. Through strategic optimization of the routing policy, as shown in Figure 2(c), we can achieve a further reduction in $C_{\mathcal{T}}$ to 1, *i.e.*, reaching the theoretically minimum communication overhead.

### 2.4. Why Collaboration Pruning?

To minimize the communication complexity $C_{\mathcal{T}}$, two potential approaches emerge:

❶ Pure system design: Dynamically optimize expert placement for each mini-batch to reduce $C_{\mathcal{T}}$.

❷ Algorithm-system co-design: Adapt the routing policy to align with a pre-defined expert placement, thereby reducing $C_{\mathcal{T}}$ while maintaining model performance.

Our empirical analysis demonstrates the unfeasibility of the pure system design approach. For a collaboration graph $\mathcal{C}$ constructed from a mini-batch $\mathcal{B}$, we analyze the number of experts in the maximal connected subgraph as the number of tokens in $\mathcal{B}$ increases. Figure 3 shows this analysis across

---

[1]The lower bound of $C_{\mathcal{T}}$ is the number of devices that $k$ can cover, which equals to $\lceil k/(N_e/N_d) \rceil = \lceil \frac{k \cdot N_d}{N_e} \rceil$.

*Figure 3.* **Empirical correlation between the expert amount in the maximal connected sub-graph and token amount in a mini-batch for top-**2**,** 4**,** 6**, and** 8 **routing**. We examine the last MoE layer for each model with the same inputs.

four MoE-LLMs with top-2, 4, 6, and 8 routing. The results show that the maximal connected subgraph converges to the complete graph as the mini-batch token count increases, implying that optimal expert partitioning across distributed devices for minimal $C_\mathcal{T}$ is practically unattainable. Therefore, we adopt an algorithm-system co-design approach to minimize the communication budget.

## 3. Method

In this section, we introduce `Occult`, an algorithm-system co-design approach that accelerates the MoE pipeline with expert parallelism by optimizing collaborative communication. Section 3.1 introduces our tailored sparse matrix multiplication (SMM) implementation for efficient all-to-all communication. Section 3.2 describes the algorithmic design for expert placement rescheduling, which critically impacts the communication budget. Section 3.3 presents our collaboration pruning schemes that push $C_\mathcal{T}$ to the limit.

### 3.1. Sparse Matrix Multiplication for Efficient All-to-All

Our system design originates from the key insights for efficient all-to-all communication. Consider 2 experts $E_i$ and $E_j$ that are co-activated by a token $x$, where the expert computing results of them must be aggregated along with $k - 2$ other experts for $x$. When $E_i$ and $E_j$ are intra-collaborated, the forward MoE pass exhibits the following properties:

* The *Dispatch* stage requires only a single replica of $x$ for all-to-all communication.

* The *Combine* stage allows pre-aggregation of expert computing results from $E_i$ and $E_j$, maintaining consistency with *Dispatch*.

* To implement this symmetric and efficient communication, the aggregation in Equation 1 should be split into two stages, *i.e.*, summing the intra-collaboration results before all-to-all, and summing the inter-collaboration results after all-to-all, for each token $x$.

To implement these ideas and enhance efficiency, we need to deal with multiple tensor states, since tokens are selectively repeated. In this paper, we outline them as 3 states:

* *Original* (`ORI`): The raw input and output tensors for an MoE layer, where each token appears exactly once.

* *Simplified* (`SFD`): The communication tensors for all-to-all operations, serving as the input of the first linear layer and the output of the second linear layer. Each token is replicated fewer than $k$ times.

* *Expanded* (`EPD`): The intermediate token tensors between the first and second linear projection layer, where each is replicated exactly $k$ times.

The token counts across states follow `ORI` < `SFD` < `EPD`. All token tensors are maintained continuously in GPU HBM. Each expert computing operator bridges two states for memory efficiency, taking one as input and producing another as output. However, direct implementation across these states presents challenges. State transitions require either token replication or reduction during computation on SRAM, necessitating token indexing. Moreover, this parallel token indexing process is still complex, due to the simultaneous consideration of devices, experts, and tokens.

To tackle this teaser, we introduce *Bidirectional Re-Index Matrix* (*BRIM*), a novel data structure for unified data management. Each MoE layer requires 2 *BRIM*s, as presented in Figure 6. A *BRIM* provides two functions, namely Scattering and Merging, which support basic MoE operations including *Dispatch*, *Combine*, and expert computing. Each function implements a two-stage process, aligned with our two-stage aggregation for Equation 1:

* Merging: Stage one (`EPD` → `SFD`) integrates with SMM, while stage two (`SFD` → `ORI`) integrates with *Dispatch* & *Combine*. As shown in the left part of Figure 5, merging operates along the $1_{st}$ dim of *BRIM*.

* Scattering: Stage one (`ORI` → `SFD`) integrates with *Dispatch* & *Combine*, while stage two (`SFD` → `EPD`) integrates with SMM. As shown in the right part of Figure 5, scattering operates along the $2_{nd}$ dim of *BRIM*.

We implement the basic MoE operations in `Occult` with Triton (Tillet et al., 2019). For *BRIM*-based SMM operators, each *thread-block* is assigned a sub-vector with a fixed number of elements in a single $BRIM_1$ row to enable tiled matrix multiplication.[2] The expert can be determined by the sub-vector's $x$-coordinate, ensuring that each *thread-block* loads

---

[2]Since *BRIM* is very sparse, directly splitting it into vectors with fixed length would cause a decrease in GPU utilization. Therefore, we further condense the obtained vectors from *BRIM* to reduce the total number of *thread-blocks* during runtime.

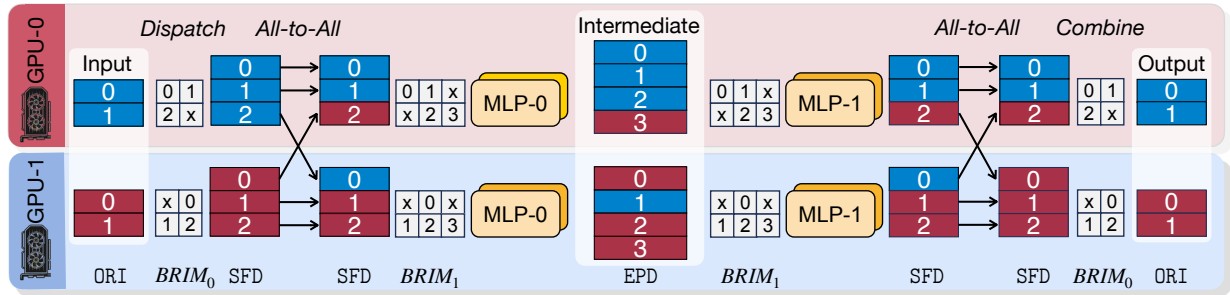

*Figure 4.* **MoE workflow with proposed bidirectional re-index-guided all-to-all communication and expert computing.** We take the forward process for illustration, where token tensors are stated as `ORI`, `SFD` and `EPD`, compactly stored on HBM and guided by *BRIM*s.

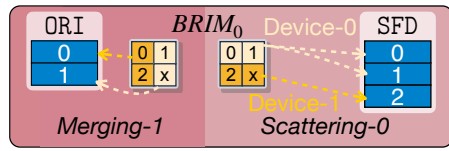

(a) Merging ($2_{nd}$ stage) and Scattering ($1_{st}$ stage) with *BRIM*$_0$

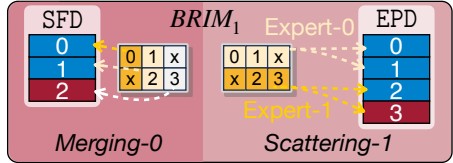

(b) Merging ($1_{st}$ stage) and Scattering ($2_{nd}$ stage) with *BRIM*$_1$

*Figure 5.* **Functions of** $2$ **BRIMs.** Both *BRIM*$_0$ and *BRIM*$_1$ are 2D matrices. *BRIM*$_0$'s dimensions are defined by the number of devices for expert parallelism (first dimension) and the token count in the `ORI` tensor (second dimension). *BRIM*$_1$'s dimensions are defined by the number of local experts per device (first dimension) and the token count in the `SFD` tensor (second dimension).

weights from one expert. Merging and scattering functions primarily differ in their I/O patterns:

★ For merging (first stage): Each *thread-block* reads from the `EPD` tensor based on the *BRIM*$_1$ sub-vector values, and writes the cumulative dot-product results to the `SFD` tensor based on the *BRIM*$_1$ vector's y-coordinates. Multiple *thread-block*s may write to the same HBM region since results from different *BRIM*$_1$ rows in the same column require aggregation (Figure 5(b)). This can be handled either through `atomic_add`[3] or by serially aggregating results across local experts, yielding identical results with similar runtime.

★ For scattering (second stage), each *thread-block* reads from the `SFD` tensor based on the *BRIM*$_1$ sub-vector's y-coordinates and writes the cumulative dot-product results to the `EPD` tensor based on the values in the *BRIM*$_1$ sub-vector.

We provide the implementation details for each basic operation in our refactored MoE workflow in Appendix A.2,

---

[3]Some data types may not be supported for `atomic_add` on NVIDIA GPUs. We initialize the output tensor with Float32, and transform it to the target type after computing.

---

**Algorithm 1 Building expert placement.**

**Require:** Collaboration graph $\mathcal{P} \in \mathbb{R}^{N_e \times N_e}$, number of devices $N_d$ for expert parallelism.
  **Initialize** expert placement $\mathcal{L}$ with $N_d$ empty lists.
  **for** $d \leftarrow 0, N_d - 1$ **do**
    **if** d == 0 **then**   ▷ Choose the 2 most collaborative experts.
      $i, j = \text{argmax}(\mathcal{P})$ and push $i, j$ to $\mathcal{L}_{[d]}$.
    **else**   ▷ Expert with the least collaboration with used ones.
      **Initialize** $\mathcal{C}_{\text{inter}}$ as an empty dict.
      **for** $e \leftarrow 0, N_e - 1$ **do**
        **if** $e$ not in $\mathcal{L}$ **then**   ▷ Traverse the unused experts.
          Set $\mathcal{C}_{\text{inter}}[e]$ as $\frac{1}{|\mathcal{L}|} \cdot \sum_{t \in \mathcal{L}} \mathcal{P}_{[t,e]}$.
        **end if**
      **end for** ▷ Expert with least collaboration with used ones.
      Index the minimal value from $\mathcal{C}_{\text{inter}}$, and push it to $\mathcal{L}_{[d]}$.
    **end if**   ▷ Initialize $\mathcal{L}_{[d]}$ with 1 or 2 experts.
    **while** len($\mathcal{L}_{[d]}$) $\leq N_e/N_d$ **do** ▷ Progressive expert selection.
      **Initialize** $\mathcal{C}_{\text{intra}}$ as an empty dict.
      **for** $e \leftarrow 0, N_e - 1$ **do**
        **if** $e$ not in $\mathcal{L}$ **then**   ▷ Traverse the unused experts.
          Set $\mathcal{C}_{\text{intra}}[e]$ as $\frac{1}{\mathcal{L}_{[d]}} \cdot \sum_{t \in \mathcal{L}_{[d]}} \mathcal{P}_{[t,e]}$.
        **end if**   ▷ Collaboration with experts on device $d$.
      **end for** ▷ Expert with the most collaboration with $\mathcal{L}_{[d]}$.
      Index the maximal value from $\mathcal{C}_{\text{intra}}$, and push it to $\mathcal{L}_{[d]}$.
    **end while**
  **end for**
  **return** $\mathcal{L}$.

---

together with a PyTorch-style pseudo-code for the whole pipeline in Appendix A.1.

### 3.2. Expert Placement Rescheduling

Since `Occult`'s communication efficiency $C_{\mathcal{T}}$ improves with increased intra-collaboration and decreased inter-collaboration, expert placement becomes critical for optimizing all-to-all communication. However, determining optimal placement is challenging due to the high variability in routing choice distributions across different mini-batches. As discussed in Section 2.4, we propose determining a fixed near-optimal expert placement for *off-the-shelf* MoE-LLMs using a profiling dataset.

We build a collaboration frequency graph $\mathcal{P}$ and formulate the rescheduling objective as evenly partitioning $\mathcal{P}$ into $N_d$ sub-graphs, maximizing intra-collaboration while minimiz-

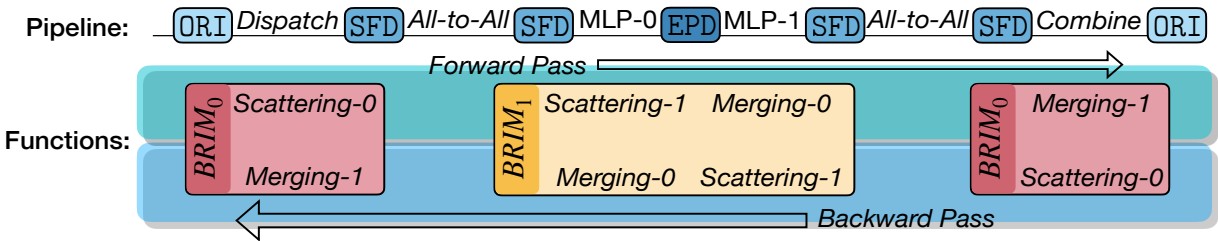

*Figure 6.* **Correlation between the functions of *BRIM* and the basic operations in MoE workflow.** Each *BRIM* provides 2 functions. *BRIM₀* is employed for *Dispatch* & *Combine*, while *BRIM₁* is adopted for the sparse matrix multiplication.

ing inter-collaboration. Each sub-graph represents device-specific expert assignments. The assigned expert placement $\mathcal{L}$ comprises $N_d$ lists. We quantify collaboration metrics:

* ⋆ Intra-collaboration for device $d$ is measured from the average collaboration frequency among all expert pairs on device $d$. With expert pair set $\mathcal{S}_d = \{(i,j)|i \in \mathcal{L}_d \land j \in \mathcal{L}_d \land i \neq j\}$:

$$\mathcal{C}^d_{\text{intra}} = \frac{1}{|\mathcal{S}_d|} \cdot \sum_{(i,j)\in\mathcal{S}_d} \mathcal{P}_{[i,j]}.$$

* ⋆ Inter-collaboration between devices $d_1$ and $d_2$ measures the average collaboration frequency across all feasible cross-device expert pairs:

$$\mathcal{C}^{d_1,d_2}_{\text{inter}} = \frac{1}{|\mathcal{L}_{d_1}| \cdot |\mathcal{L}_{d_2}|} \cdot \sum_{i\in\mathcal{L}_{d_1}, j\in\mathcal{L}_{d_2}} \mathcal{P}_{[i,j]}.$$

To derive a near-optimal expert allocation scheme from $\mathcal{P}$, we develop a clustering algorithm in Algorithm 1, inspired by farthest point sampling in point cloud learning (Qi et al., 2017). Compared to trivial expert placement, our rescheduling algorithm reduces $C_{\mathcal{T}}$ by $\sim 20\%$ while maintaining exact computation results, as shown in Table 1.

### 3.3. Expert Collaboration Pruning

We further optimize collaborative communication through algorithm-level collaboration pruning. This approach restricts the routing choice of each token to a pre-defined range of $N_d$ devices. Collaborations outside this scope are pruned and replaced by a legal alternative. The $N_d$ devices are determined by traversing the tokens' $k$ routed experts in descending order of routing score until $N_d$ devices are reached. [4] Expert replacement follows two proposed criteria for pruning implementation.

**Routing-Score-based Pruning.** We start by getting the gating scores for all experts as the router network output, denoted as $\{r(\boldsymbol{x})_i\}_{i=0}^{N_e-1}$. While conventional top-$k$ routing selects the largest $k$ values for forward pass and weighted aggregation (Equations 1 and 2), we modify it with 2 steps:

❶ Gating score sorting: Sort $\{r(\boldsymbol{x})_i\}_{i=0}^{N_e-1}$ as $\{r(\boldsymbol{x})'_i\}_{i=0}^{N_e-1}$ in descending order.

❷ Expert selection: Select top-$k$ experts within the range of the $N_d$ devices based on $\{r(\boldsymbol{x})'_i\}_{i=0}^{N_e-1}$.

---

[4]Pruning would not be required if the $k$ routed experts cover less than $N_d$ devices

**Expert-Similarity-based Pruning.** We prepare an expert similarity table for each MoE layer, named as $\mathcal{T} \in \mathbb{Z}^{N_e \times N_e}$, and then traverse the $k$ routed experts in descending order of routing score. Experts outside the $N_d$ devices range are replaced with the most similar available alternative from $\mathcal{T}$ that satisfies two conditions: (1) falls within the range of $N_d$ devices, and (2) has not been selected before.

Following the empirical insights from MC-MoE (Li et al., 2023b), we measure the expert similarity using router logits obtained from the inference process on a profiling dataset. Implementation details are provided in Appendix A.3.

## 4. Experiments

### 4.1. Experimental Setup

**Models, Tasks, and Datasets.** Models: We examine the effectiveness of Occult on three frontier MoE-LLMs: OLMoE-1B-7B (Muennighoff et al., 2024), Qwen1.5-MoE-A2.7B (Team, 2024), and DeepSeek-MoE (Dai et al., 2024). The configurations are provided in Table 2. Tasks and datasets: To validate the effectiveness of collaboration pruning, we tune the models with Alpaca (Taori et al., 2023), and evaluate on six tasks: Winogrande (Sakaguchi et al., 2021), WSC (Levesque et al., 2012), PIQA (Bisk et al., 2020), RACE (Lai et al., 2017), MathQA (Amini et al., 2019) and RTE (Bentivogli et al., 2009). A more comprehensive evaluation with extensive benchmarks is provided in Appendix B.

**Baselines and Evaluation Metrics.** Baselines: For inference, we compare against vllm (Kwon et al., 2023) and HuggingFace. For training, we compare against PyTorch fully-sharded data parallel (FSDP) and DeepSpeed (Rajbhandari et al., 2022). We evaluate two popular MoE libraries: Tutel (Hwang et al., 2023) and MegaBlocks (Gale et al., 2023) (For MegaBlocks, both block sparse matrix multiplication and grouped GeMM for dMoE are examined). Since Occult is orthogonal to these general frameworks, vllm and HuggingFace serve as references only. Evaluation metrics: We report accuracy for NLP tasks involved in this section, with F1 score and exact match score results presented in Appendix B.

**Hardware and Software.** All the experiments are conducted on a single node using PCIe-connected NVIDIA

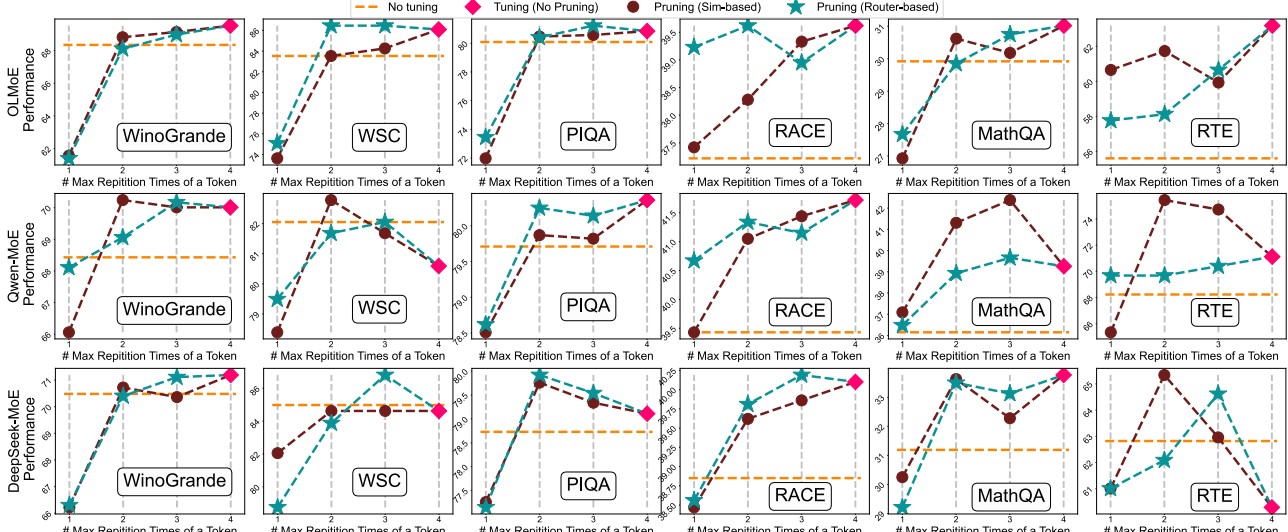

*Figure 7.* **Performance Comparison for Collaboration Pruning.** Comprehensive evaluation across three MoE architectures shows performance trends under different pruning strategies. Note that 4-device collaboration pruning is equivalent to standard training with original top-k routing.

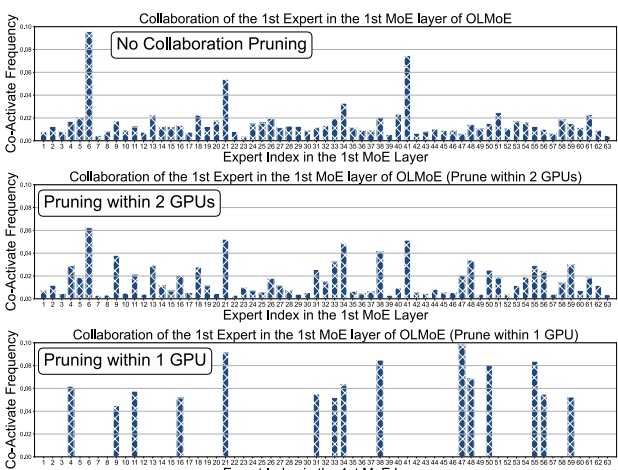

*Figure 8.* **Collaboration Analysis for OLMoE using 4-way expert parallelism**. Examining expert 0 in the first MoE layer shows stable collaboration between no-pruning and 2-GPU scenarios, while 1-GPU pruning leads to some patterns' vanishing.

A6000 GPUs, each with 48GB HBM memory. We take BFloat16 for MoE parameters and activations, and Occult implements atomic_add operations in Float32. Batch size and token amounts are recorded for individual devices.

### 4.2. Performance of Collaboration Pruning

**Effectiveness of Collaboration Pruning.** Figure 7 presents performance comparisons for both router- and similarity-based pruning strategies with various device amounts. Our results demonstrate that:

❶ While restricting collaboration within 1 device may hamper the model performance, expanding the constraint to 2 devices can achieve comparable or superior quality than standard training.

❷ Although router-based pruning is easier to implement, similarity-based pruning consistently outperforms it.

**Explanation for Performance Enhancement.** To interpret the collaboration mechanism, we visualize an expert from OLMoE's first MoE layer in Figure 8. Single-device pruning only preserves certain correlations while others are wiped out. In contrast, two-device pruning can ensure the potential collaboration between any pair of experts, maintaining correlation patterns that more closely align with the original model's structure.

### 4.3. Accelerated Expert Parallelism

**Prefilling.** Figure 9 demonstrates Occult's superior latency performance compared to existing frameworks on communication-intensive tasks. Using $2^{14}$ prompt tokens with Occult, pruning on two devices, as our benchmark:

★ For OLMoE, it is $8.66\times$ faster than Tutel, $1.70\times$ faster than MegaBlocks and $1.86\times$ faster than vllm.

★ For Qwen-MoE, it is $2.74\times$ faster than Tutel, $0.71\times$ faster than MegaBlocks and $1.60\times$ faster than vllm.

★ For DeepSeek-MoE, it is $5.64\times$ faster than Tutel, $0.66\times$ faster than MegaBlocks and $1.51\times$ than vllm.

Pruning within 1 device can achieve the highest efficiency, but with slightly inferior quality, while pruning within 2 devices can achieve a superior trade-off.

**Decoding.** We visualize the latency comparison for full generation in Figure 10. While Occult excels with large batch generation and demonstrates the most consistent scaling across MoE-LLMs, indicating superior throughput, its acceleration is limited for small batch sizes where communication is not necessarily a bottleneck.

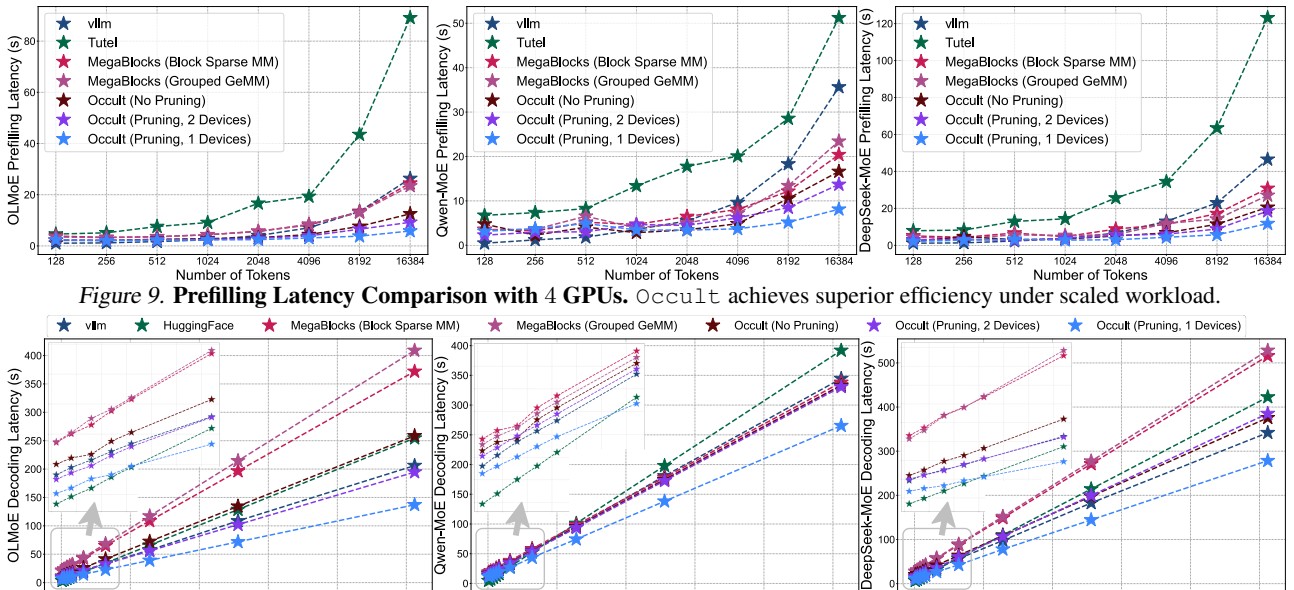

*Figure 9.* **Prefilling Latency Comparison with** 4 **GPUs.** Occult achieves superior efficiency under scaled workload.

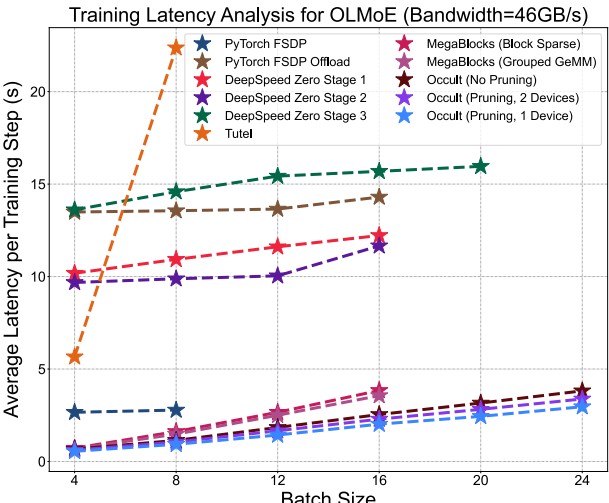

*Figure 10.* **Decoding Latency Comparison with** 4 **GPUs.** Analysis with fixed prompt tokens (12800) and batch size (512) demonstrates Occult's consistent latency advantages on communication-intensive decoding tasks.

*Figure 11.* **Training Latency Comparison with** 4 **GPUs.** With fixed sequence length of 512 tokens, Occult demonstrates superior performance across varying batch sizes.

**Training.** We evaluate the average latency per training step over 1k iterations with sequence length 512. We evaluate on OLMoE (Muennighoff et al., 2024) and DeepSeek-MoE (Dai et al., 2024) through tuning all the MoE modules and freezing other parameters. We take a comprehensive comparison among popular training frameworks with 4 GPUs in Figure 11, and examine the expert parallelism training frameworks with 8 and 16 GPUs in Figure 12. Experimental results demonstrate that Occult possesses superior memory efficiency, supporting the largest batch size under the same memory budget. Occult can also speed up fine-tuning for MoE-based LLMs. For batch size 8, Occult (two-device pruning) achieves $1.54\times$, $2.65\times$,

$\sim 9\times$, and $\sim 20\times$ speedup compared to MegaBlocks (Gale et al., 2023), FSDP (PyTorch Fully Sharded Data Parallelism (Zhao et al., 2023)), DeepSeed (Rajbhandari et al., 2022), and Tutel (Hwang et al., 2023), respectively. Occult is scalable for increasing GPUs involved in expert parallelism, while exhibiting the most stable latency increasing trend with batch size growth compared to other expert-parallel frameworks.

## 5. Related Works

**Open-Source MoE-based LLMs.** Most of the modern MoE-LLMs adopt top-$k$ routing with $k \geq 2$, implying that collaborative communication is universal. This enables Occult as a general approach to advance efficiency for scalable expert parallelism. We investigate some frontier open-source MoE-LLMs in Table 2. The attempts in the early stage for sparsely scaling language models include Nllb (Costa-jussà et al., 2022) and Switch-Transformers (Fedus et al., 2022). In recent years, we have witnessed the vibrant development of modern MoE-LLMs. For instance, Mixtral (Jiang et al., 2024) outperforms Llama2-70B (Touvron et al., 2023) with $8 \times 7$B parameters. Phi-3.5-MoE (Abdin et al., 2024) contains $16 \times 3.8$B parameters, which is on par with Gemini-1.5-Flash and GPT-4o-mini. Qwen1.5-MoE (Team, 2024) activates 2.7B parameters and matches the performance of 7B models. DeepSeek-MoE (Dai et al., 2024) introduces shared experts to learn common knowledge. OLMoE (Muennighoff et al., 2024) open-sources all data related to model training. PowerMoE (Shen et al., 2024b) trains the MoE model with a new learning rate scheduler. JetMoE (Shen et al., 2024a) surpasses Llama2-7B (Touvron et al., 2023) with low training cost.

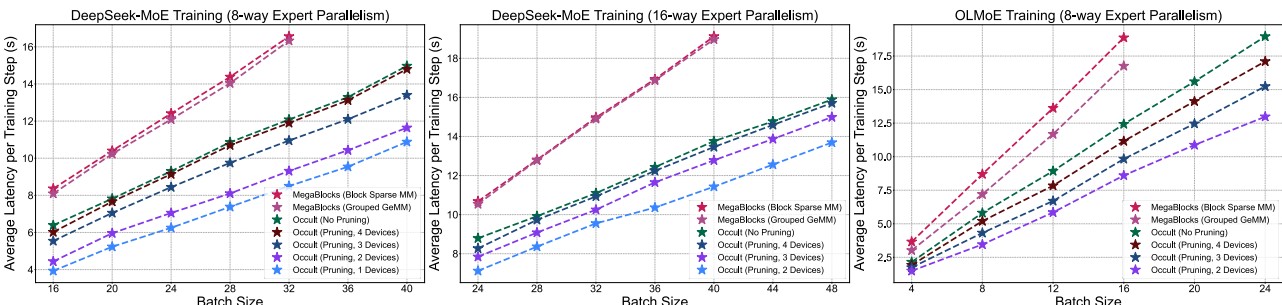

*Figure 12.* **More training latency comparison for expert parallelism frameworks.** Owning to the communication- and memory-efficient design, `Occult` achieves superior training efficiency under both 8- and 16-way expert parallelism configurations.

*Table 2.* **Frontier Open-Source MoE-LLMs.** The models we used for experiments are highlighted in bold.

| Model | $k$ | # Experts | # MoE Layers | # Params |
|---|---|---|---|---|
| Mixtral-8x7B (Jiang et al., 2024) | 2 | 8 | 32 | 46.7 B |
| Mixtral-8x22B (Jiang et al., 2024) | 2 | 8 | 56 | 141 B |
| Phi-3.5-MoE (Abdin et al., 2024) | 2 | 16 | 32 | 41.9 B |
| Minimax-01 (Li et al., 2025) | 2 | 32 | 80 | 456 B |
| **Qwen1.5-MoE (Team, 2024)** | 4 | 60 | 24 | 14.3 B |
| **DeepSeek-MoE (Dai et al., 2024)** | 6 | 64 | 27 | 16.4 B |
| DeepSeek-V2 (Liu et al., 2024a) | 6 | 160 | 59 | 236 B |
| **OLMoE (Muennighoff et al., 2024)** | 8 | 64 | 16 | 6.92 B |
| Qwen3-30B-A3B (Team, 2025) | 8 | 128 | 48 | 30.5 B |
| Qwen3-235B-A22B (Team, 2025) | 8 | 128 | 94 | 235 B |
| DeepSeek-V3 (Liu et al., 2024b) | 8 | 256 | 58 | 685 B |

**System Designs for Efficient MoE.** DeepSpeed-MoE (Rajbhandari et al., 2022) designs a new MoE architecture and an optimized inference system for efficient and scalable serving. Tutel (Hwang et al., 2023) provides a scalable MoE framework with adaptive parallelism/pipelining optimization at runtime. MegaBlocks (Gale et al., 2023) proposes block-sparse matrix multiplication to enable no discard of tokens. Hexa-MoE (Luo et al., 2024) proposes expert-specific operators as an alternative to GeMM or grouped GeMM to make MoE-training heterogeneous-aware. Janus (Liu et al., 2023) proposes a data-centric strategy to eliminate all-to-all communication overhead under a large workload. APTMoE (Wei et al., 2024) proposes an affinity-aware offloading technique to enhance pipeline parallelism for fine-tuning MoE-LLMs.

**Algorithm Designs for Efficient MoE.** Hash-Layer (Roller et al., 2021) replaces the gating layer in the common MoE model with a precomputed hash function to reduce computation cost. MoE-$I^2$ (Yang et al., 2024) proposes a two-stage compression scheme, including inter-expert pruning and intra-expert low-rank decomposition. Pre-gated MoE (Hwang et al., 2024) proposes a pre-gating function to enable the pre-fetching of MoE parameters so that it can be efficiently served on memory-constrained devices. Expert Pruning (Lu et al., 2024) proposes post-training approaches for task-agnostic and task-specific expert pruning and skipping with MoE-LLMs to reduce the model size and improve inference

efficiency. MC-MoE (Li et al., 2023b) merges the experts into low-rank and structurally sparse alternatives for better efficiency of inference.

**Principles of GPU Acceleration.** Modern GPUs provide massive threads for parallel execution. Threads are grouped into *thread-block*s, which are executed on streaming multi-processors (SMs). GPUs have a memory hierarchy, outlined as large but slow-accessed high bandwidth memory (HBM) and small but faster-accessed shared memory (SRAM). Matrix multiplication is optimized on GPU using *tiling*, *i.e.*, partitioning the output matrix into small 2D blocks, where each block is computed using a *thread-block* in parallel. The size of an individual block can be manually adjusted to improve runtime performance.

# 6. Conclusion

In this paper, we introduce `Occult`, an algorithm-system co-design approach to optimize all-to-all communication in expert parallelism for accelerated MoE training and inference. Comprehensive experiments show that it can consistently speed up communication-intensive tasks for MoE-LLMs. `Occult` is orthogonal to popular frameworks, therefore, it can be further integrated with them for enhanced efficiency. As preliminary research to advance efficiency in expert parallelism, some problems remain unsolved, such as workload balancing. We will continue to explore this topic and try to provide improved solutions.

# Acknowledgements

Pingzhi Li and Tianlong Chen are partially supported by Amazon Research Award, Cisco Faculty Award, UNC Accelerating AI Awards, NAIRR Pilot Award, OpenAI Researcher Access Award, and Gemma Academic Program GCP Credit Award.

# Impact Statement

This paper aims to improve the communication efficiency for training and inference with modern MoE-based LLMs. The efficiency advantage of such models might help democratize access of language models. On the other hand, whether such new algorithm would affect known issues such as biased and harmful outputs of language models remains an unexplored research question.

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

# A. Algorithm Details.

## A.1. Refactored MoE Pipeline

We demonstrate the refactored MoE pipeline in Algorithm 2. Notice that the routing weights for the top-$k$ experts are modulated in the intermediate tokens rather than the output to ensure a symmetric pipeline. We will provide the details for each basic MoE operator in the following.

---

**Algorithm 2 PyTorch-style pseudocode of MoE pipeline.**

```
1  # x.shape: (num_tokens_ori, hidden_size)
2  def MoE_forward(self, x):
3      # (1) Assign tokens to experts.
4      # ids.shape: (num_tokens_ori, self.top_k), w.shape: (num_tokens_ori, self.top_k)
5      ids, w = router(x)
6
7      # (2) Build BRIM-0 for dispatch & combine.
8      # brim_0.shape: (num_devices, num_tokens_ori)
9      brim_0 = build_brim_0(x, ids)
10
11     # (3) Dispatch
12     # x.shape: (num_tokens_sfd_0, hidden_size), ids.shape: (num_tokens_sfd_0, self.top_k)
13     # w.shape: (num_tokens_sfd_0, self.top_k)
14     x, ids, w = dispatch(x, ids, w, brim_0)
15
16     # (4) 1st all-to-all communication
17     # x.shape: (num_tokens_sfd_1, hidden_size), ids_sfd.shape: (num_tokens_sfd_1, self.top_k)
18     # w_sfd: (num_tokens_sfd_1, self.top_k)
19     x, ids_sfd, w_sfd = all_to_all_0(x, ids, w, brim_0)
20
21     # (5) Build BRIM-1 for expert computing
22     # brim_1.shape: (num_local_exps, num_tokens_sfd_1)
23     # brim_w.shape: (num_local_exps, num_tokens_sfd_1)
24     brim_1, brim_w = build_brim_1(x, ids_sfd, w_sfd)
25
26     # (6) Computing with local experts
27     # x_epd.shape: (num_tokens_epd, intermediate_size), x.shape: (num_tokens_sfd_1, hidden_size)
28     x_epd = smm_scattering(x, self.w1, brim_1)
29     x_epd = weight_modulate(x_epd, brim_1, brim_w)
30     x = smm_merging(x_epd, self.w2, brim_1)
31
32     # (7) 2nd all-to-all communication
33     # x.shape: (num_tokens_sfd_0, hidden_size)
34     x = all_to_all_1(x, brim_1)
35
36     # (8) Combine
37     # x.shape: (num_tokens_ori, hidden_size)
38     x = combine(x, brim_0)
39
40     return x
```

`weight_modulate`: assign weights to tokens with `EPD` state in parallel.

---

## A.2. Algorithms Details for Basic MoE Operations

**Build $BRIM_0$.** $BRIM_0$ is constructed on each device before *Dispatch* as an guidance. We provide the details in Algorithm 3.

---

**Algorithm 3 Building $BRIM_0$.**

**Require:** Input tokens $x$ shaped as $(N_{ori}, D)$, routing choice $r$ shaped as $(N_{ori}, N_e)$, number of devices $N_d$ for expert parallel, expert placement $\mathcal{L}$ shaped as $(N_d, \lceil N_e/N_d \rceil)$.

    **Initialize** $BRIM_0$ with an empty tensor shaped as $(N_d, N_{ori})$.
    **Initialize** $ctr \leftarrow 0$.
    **for** $d \leftarrow 0, N_d - 1$ **do**
        **for** $t \leftarrow 0, N_{ori} - 1$ **do**
            **if** token $x_{t,*}$ activates experts kept on device $d$ **then**
                $BRIM_0[d,t] \leftarrow ctr$.
                $ctr \leftarrow ctr + 1$
            **else**
                $BRIM_0[d,t] \leftarrow -1$.
            **end if**
        **end for**
    **end for**
    **return** $BRIM_0$.

---

***Dispatch.*** In *Dispatch* stage, tokens, routing choices, and routing weights are processed together to selectively repeat for each token. We provide the algorithm details in Algorithm 4, where only tokens are processed as an example.

---

**Algorithm 4 Dispatch.**

---

**Require:** Input tokens $x$ shaped as $(N_{\text{ori}}, D)$, *BRIM*$_0$ shaped as $(N_{\text{d}}, N_{\text{ori}})$, tile size $(\texttt{M}, \texttt{N})$.
    **Initialize** $N_{\text{sfd}}$ with the amount of non-negative elements in *BRIM*$_0$.
    **Initialize** $x_{\text{sfd}}$ with an empty tensor shaped as $(N_{\text{sfd}}, D)$.
    **Prepare** $N_{\text{d}} \cdot \lceil N_{\text{ori}}/\texttt{M} \rceil \cdot \lceil D/\texttt{N} \rceil$ *thread-block*s
    **parfor** $d \leftarrow 0, N_{\text{d}} - 1$ **do**
        **parfor** $t_{\text{m}} \leftarrow 0, \lceil N_{\text{ori}}/\texttt{M} \rceil - 1$ **do**
            **parfor** $t_{\text{n}} \leftarrow 0, \lceil D/\texttt{N} \rceil - 1$ **do**
                Load *BRIM*$_0[d, t_{\text{m}} \cdot \texttt{M} : (t_{\text{m}} + 1) \cdot \texttt{M}]$ from HBM to SRAM, denoted as $v$.
                Load $x[t_{\text{m}} \cdot \texttt{M} : (t_{\text{m}} + 1) \cdot \texttt{M}, t_{\text{n}} \cdot \texttt{N} : (t_{\text{n}} + 1) \cdot \texttt{N}]$ from HBM to SRAM, denoted as $c$.
                Write $c$ to $x_{\text{sfd}}[v, t_{\text{n}} \cdot \texttt{N} : (t_{\text{n}} + 1) \cdot \texttt{N}]$ from SRAM to HBM.
            **end parfor**
        **end parfor**
    **end parfor**
    **return** $x_{\text{sfd}}$.

---

**All-to-All.** We take the `all_to_all_single` interface of `torch.distributed` to implement the all-to-all communication in `Occult`. On each device, the tokens after *Dispatch* can be further transmitted to device $d$ based on the non-zero elements in *BRIM*$_0[d, *]$, which serve as the indices.

**Build *BRIM*$_1$.** *BRIM*$_1$ is constructed after the 1st all-to-all communication to guide the sparse matrix multiplication kernels. The details for construction are provided in Algorithm 5.

---

**Algorithm 5 Building *BRIM*$_1$.**

---

**Require:** Input tokens $x_{\text{sfd}}$ shaped as $(N_{\text{sfd}}, D)$, routing choice $r_{\text{sfd}}$ shaped as $(N_{\text{sfd}}, N_{\text{locexp}})$, local expert list $\mathcal{L}_{\text{loc}}$ shaped as $(N_{\text{locexp}}, )$.
    **Initialize** *BRIM*$_1$ with an empty tensor shaped as $(N_{\text{locexp}}, N_{\text{sfd}})$.
    **Initialize** $ctr \leftarrow 0$.
    **for** $e \leftarrow 0, N_{\text{locexp}} - 1$ **do**
        **for** $t \leftarrow 0, N_{\text{sfd}} - 1$ **do**
            **if** token $x_{\text{sfd}}[t, *]$ activates expert $\mathcal{L}_{\text{loc}}[e]$ **then**
                *BRIM*$_1[e, t] \leftarrow ctr$
                $ctr \leftarrow ctr + 1$
            **else**
                *BRIM*$_1[e, t] \leftarrow -1$
            **end if**
        **end for**
    **end for**
    **return** *BRIM*$_1$.

---

**Sparse Matrix Multiplication (Scattering).** Sparse matrix multiplication with scattering function takes a token tensor with `SFD` state and weights as input, and a token tensor with `EPD` state as output, where the weight tensor is dense while the others are sparse. Details are provided in Algorithm 6.

**Algorithm 6 Sparse Matrix Multiplication with Scattering.**

**Require:** Input tokens $x_{\text{sfd}}$ shaped as $(N_{\text{sfd}}, D_{\text{in}})$, expert weights $w$ shaped as $(N_{\text{locexp}}, D_{\text{in}}, D_{\text{out}})$, $BRIM_1$ shaped as $(N_{\text{locexp}}, N_{\text{sfd}})$, tile size $(\texttt{M}, \texttt{K}, \texttt{N})$.
    **Initialize** output tokens $x_{\text{epd}}$ with an empty tensor shaped as $(N_{\text{epd}}, D_{\text{out}})$.
    **Prepare** $N_{\text{locexp}} \cdot \lceil N_{\text{sfd}}/\texttt{M} \rceil \cdot \lceil D_{\text{out}}/\texttt{N} \rceil$ *thread-block*s.
    **parfor** $e \leftarrow 0, N_{\text{locexp}} - 1$ **do**
        **parfor** $t_{\text{m}} \leftarrow 0, \lceil N_{\text{sfd}}/\texttt{M} \rceil - 1$ **do**
            **parfor** $t_{\text{n}} \leftarrow 0, \lceil D_{\text{out}}/\texttt{N} \rceil - 1$ **do**
                On chip: **Initialize** $c = 0 \in \mathbb{R}^{\texttt{M} \times \texttt{N}}$
                **for** $k \leftarrow 0, \lceil D_{\text{in}}/\texttt{K} \rceil - 1$ **do**
                    Load $x_{\text{sfd}}[t_{\text{m}} \cdot \texttt{M} : (t_{\text{m}} + 1) \cdot \texttt{M}, k \cdot \texttt{K} : (k + 1) \cdot \texttt{K}]$ from HBM to SRAM as $a$.
                    Load $w[e, k \cdot \texttt{K} : (k + 1) \cdot \texttt{K}, t_{\text{n}} \cdot \texttt{N} : (t_{\text{n}} + 1) \cdot \texttt{N}]$ from HBM to SRAM as $b$.
                    On chip: $c \leftarrow c + a \cdot b$.
                **end for**
                Load $BRIM_1[e, t_{\text{m}} \cdot \texttt{M} : (t_{\text{m}} + 1) \cdot \texttt{M}]$ from HBM to SRAM, denoted as $v$.
                Write $c$ to $x_{\text{epd}}[v, t_{\text{n}} \cdot \texttt{N} : (t_{\text{n}} + 1) \cdot \texttt{N}]$ from SRAM to HBM.
            **end parfor**
        **end parfor**
    **end parfor**
    **return** $x_{\text{epd}}$.

**Weight Modulation.** Since the MoE pipeline of `Occult` is symmetric, we modulate the routing weights on the intermediate tokens rather than the output, which delivers the same results. Details are provided in Algorithm 7.

**Algorithm 7 Weight Modulation.**

**Require:** Intermediate tokens $x_{\text{epd}}$ shaped as $(N_{\text{epd}}, D)$, routing weights $w$ shaped as $(N_{\text{sfd}}, N_{\text{e}})$, $BRIM_1$ shaped as $(N_{\text{locexp}}, N_{\text{sfd}})$, local expert list $\mathcal{L}_{\text{loc}}$ shaped as $(N_{\text{locexp}}, )$, tile size $(\texttt{N}, )$.
    **Initialize** modulated tokens $x'_{\text{epd}}$ with empty tensor shaped as $(N_{\text{epd}}, D)$.
    **Prepare** $N_{\text{locexp}} \cdot N_{\text{sfd}} \cdot \lceil D/\texttt{N} \rceil$ *thread-block*s.
    **parfor** $e \leftarrow 0, N_{\text{locexp}} - 1$ **do**
        **parfor** $t \leftarrow 0, N_{\text{sfd}} - 1$ **do**
            **parfor** $t_{\text{n}} \leftarrow 0, \lceil D/\texttt{N} \rceil - 1$ **do**
                **if** $BRIM_1[e, t] \geq 0$ **then**
                    Load $x_{\text{epd}}[BRIM_1[e, t], t_{\text{n}} \cdot \texttt{N} : (t_{\text{n}} + 1) \cdot \texttt{N}]$ from HBM to SRAM, denoted as $c$.
                    Write $c \cdot w[t, \mathcal{L}_{\text{loc}}[e]]$ to $x'_{\text{epd}}[BRIM_1[e, t], t_{\text{n}} \cdot \texttt{N} : (t_{\text{n}} + 1) \cdot \texttt{N}]$ from SRAM to HBM.
                **end if**
            **end parfor**
        **end parfor**
    **end parfor**
    **return** $x'_{\text{epd}}$.

**Sparse Matrix Multiplication (Merging).** Sparse Matrix Multiplication with merging function takes a token tensor with `EPD` state and weights as input, and a token tensor with `SFD` state as output, where the weight tensor is dense while the others are sparse. Details are provided in Algorithm 8.

**Algorithm 8 Sparse Matrix Multiplication with Merging.**

---

**Require:** Input tokens $x_{\text{epd}}$ shaped as $(N_{\text{epd}}, D_{\text{in}})$, expert weights $w$ shaped as $(N_{\text{locexp}}, D_{\text{in}}, D_{\text{out}})$, $BRIM_1$ shaped as $(N_{\text{locexp}}, N_{\text{sfd}})$, tile size $(\texttt{M}, \texttt{K}, \texttt{N})$.
    **Initialize** output tokens $x_{\text{sfd}}$ with a zero tensor shaped as $(N_{\text{sfd}}, D_{\text{out}})$.
    **Prepare** $N_{\text{locexp}} \cdot \lceil N_{\text{sfd}}/\texttt{M} \rceil \cdot \lceil D_{\text{out}}/\texttt{N} \rceil$ *thread-block*s.
    **parfor** $e \leftarrow 0, N_{\text{locexp}} - 1$ **do**
        **parfor** $t_{\text{m}} \leftarrow 0, \lceil N_{\text{sfd}}/\texttt{M} \rceil - 1$ **do**
            **parfor** $t_{\text{n}} \leftarrow 0, \lceil D_{\text{out}}/\texttt{N} \rceil - 1$ **do**
                On chip: **Initialize** $c = 0 \in \mathbb{R}^{\texttt{M} \times \texttt{N}}$.
                Load $BRIM_1[e, t_{\text{m}} \cdot \texttt{M} : (t_{\text{m}} + 1) \cdot \texttt{M}]$ from HBM to SRAM, denoted as $v$.
                **for** $k \leftarrow 0, \lceil D_{\text{in}}/\texttt{K} \rceil - 1$ **do**
                    Load $x_{\text{epd}}[v, k \cdot \texttt{K} : (k + 1) \cdot \texttt{K}]$ from HBM to SRAM as $a$.
                    Load $w[e, k \cdot \texttt{K} : (k + 1) \cdot \texttt{K}, t_{\text{n}} \cdot \texttt{N} : (t_{\text{n}} + 1) \cdot \texttt{N}]$ from HBM to SRAM as $b$.
                    On chip: $c \leftarrow c + a \cdot b$.
                **end for**
                Write $c$ to $x_{\text{sfd}}[t_{\text{m}} \cdot \texttt{M} : (t_{\text{m}} + 1) \cdot \texttt{M}, t_{\text{n}} \cdot \texttt{N} : (t_{\text{n}} + 1) \cdot \texttt{N}]$ from SRAM to HBM via $\texttt{Atomic\_Add}$.
            **end parfor**
        **end parfor**
    **end parfor**
    **return** $x_{\text{sfd}}$.

---

***Combine*.** In *Combine* stage, different replicas of the same token computed on different devices are merged into the final output of an MoE layer. Details are provided in Algorithm 9.

**Algorithm 9 Combine.**

---

**Require:** Input tokens $x_{\text{sfd}}$ shaped as $(N_{\text{sfd}}, D)$, $BRIM_0$ shaped as $(N_{\text{d}}, N_{\text{ori}})$, tile size $(\texttt{M}, \texttt{N})$.
    **Initialize** $x_{\text{ori}}$ with a zero tensor shaped as $(N_{\text{ori}}, D)$.
    **for** $d \leftarrow 0, N_{\text{d}} - 1$ **do**
        **parfor** $t_{\text{m}} \leftarrow 0, \lceil N_{\text{ori}} \rceil - 1$ **do**
            **parfor** $t_{\text{n}} \leftarrow 0, \lceil D \rceil - 1$ **do**
                Load $BRIM_0[d, t_{\text{m}} \cdot \texttt{M} : (t_{\text{m}} + 1) \cdot \texttt{M}]$ from HBM to SRAM, denoted as $v$.
                Load $x_{\text{sfd}}[v, t_{\text{n}} \cdot \texttt{N} : (t_{\text{n}} + 1) \cdot \texttt{N}]$ from HBM to SRAM, denoted as $c$.
                Add $c$ to $x_{\text{ori}}[t_{\text{m}} \cdot \texttt{M} : (t_{\text{m}} + 1) \cdot \texttt{M}, t_{\text{n}} \cdot \texttt{N} : (t_{\text{n}} + 1) \cdot \texttt{N}]$ from SRAM to HBM.
            **end parfor**
        **end parfor**
    **end for**
    **return** $x_{\text{ori}}$.

---

### A.3. Similarity Table Construction

The empirical insights of MC-MoE (Li et al., 2023b) inspire us to measure the expert similarity with router logits, *i.e.*, experts with similar router logits tend to be more interchangeable with each other. A profiling dataset is required in this way. We denote the profiled router logits from model inference as $\texttt{H} \in \mathbb{R}^{N_{\text{t}} \cdot N_{\text{e}}}$, including $N_{\text{ori}}$ tokens and $N_{\text{e}}$ experts. Similarity table $\mathcal{T} \in \mathbb{Z}^{N_{\text{e}} \times N_{\text{e}}}$ can be constructed via the cosine similarity between $\texttt{H}^{\top}$ and $\texttt{H}$:

$$\mathcal{T}_{i,j} = \frac{\langle \texttt{H}^{\top}_{*,i}, \texttt{H}_{*,j} \rangle}{\|\texttt{H}_{*,i}\|_2 \cdot \|\texttt{H}_{*,j}\|_2}, \tag{6}$$

To make the similarity table more representative, the profiled token amount should be large enough. However, directly compute cosine similarity for it would cause overflow for floating point number. Therefore we change to compute the squared similarity:

$$\mathcal{T}^2_{i,j} = \frac{\langle \texttt{H}^{\top}_{*,i}, \texttt{H}_{*,j} \rangle^2}{\|\texttt{H}_{*,i}\|^2_2 \cdot \|\texttt{H}_{*,j}\|^2_2}, \tag{7}$$

We further take an average for both numerator and denominator to normalize their scope:

$$\mathcal{T}^2_{i,j} = \frac{\frac{1}{N_{\text{ori}}} \cdot \langle \texttt{H}^{\top}_{*,i} \cdot \texttt{H}_{*,j} \rangle^2}{\frac{1}{N_{\text{ori}}} \cdot \|\texttt{H}_{*,i}\|^2_2 \cdot \|\texttt{H}_{*,j}\|^2_2}, \tag{8}$$

This can be computed via accumulating small batches for both numerator and denominator.

## B. Full Evaluation Results

We implement comprehensive evaluation to validate the effectiveness of collaboration pruning on model performance with 23 popular benchmarks.

For OLMoE, we provide the results in Table 3, where the raw *off-the-shelf* model performs best on 5 benchmarks. Apart from these tasks, router-based pruning outperforms similarity-based pruning on 12 benchmarks in total, while similarity-based pruning outperforms router-based pruning on 8 benchmarks.

*Table 3.* **Full Evaluation Results for Collaboration Pruning with OLMoE.**

| Method | Metric | Raw | Router-based Prune, 1 GPU | Router-based Prune, 2 GPUs | Sim-based Prune, 1 GPU | Sim-based Prune, 2 GPUs | Standard SFT | Router-based Better | Sim-based Better |
|---|---|---|---|---|---|---|---|---|---|
| WSC (Levesque et al., 2012) | accuracy | 83.52 | 75.09 | **86.45** | 73.63 | 83.52 | 86.08 | ✓ | |
| Winogrande (Sakaguchi et al., 2021) | accuracy | 68.35 | 61.40 | 68.11 | 61.56 | 68.82 | **69.53** | | ✓ |
| ASDiv (Miao et al., 2020) | accuracy | 4.56 | 2.86 | **9.59** | 2.86 | 8.33 | 9.37 | ✓ | |
| OpenBookQA (Mihaylov et al., 2018) | accuracy | 33.20 | 26.00 | **34.60** | 23.60 | 32.40 | 34.20 | ✓ | |
| PIQA (Bisk et al., 2020) | accuracy | 80.09 | 73.45 | 80.41 | 71.98 | 80.47 | **80.85** | | ✓ |
| HellaSwag (Zellers et al., 2019) | accuracy | 58.07 | 48.14 | 57.53 | 47.14 | 56.99 | **59.91** | ✓ | |
| SST-2 (Socher et al., 2013) | accuracy | 62.61 | 75.92 | **86.35** | 74.66 | 85.44 | 83.94 | ✓ | |
| MultiNLI (Williams et al., 2018) | accuracy | 40.96 | 35.82 | 41.93 | 33.64 | 43.23 | **45.04** | | ✓ |
| QASPER (Dasigi et al., 2021) | F1 score | 90.53 | 93.88 | 97.03 | 97.03 | 97.03 | **97.79** | | ✓ |
| MRPC (Dolan & Brockett, 2005) | accuracy | 51.96 | 66.18 | 67.65 | **68.63** | 68.14 | 65.93 | | ✓ |
| MRPC (Dolan & Brockett, 2005) | F1 score | 59.84 | 79.09 | 80.65 | **81.34** | 81.05 | 79.35 | | ✓ |
| MultiRC (Khashabi et al., 2018) | accuracy | 57.16 | 57.22 | **57.24** | 56.70 | 57.10 | 57.16 | ✓ | |
| WNLI (Wang et al., 2018) | accuracy | 54.93 | 53.52 | **60.56** | 59.15 | 57.75 | 53.52 | ✓ | |
| RTE (Bentivogli et al., 2009) | accuracy | 55.60 | 57.76 | 58.12 | 60.65 | 61.73 | **63.18** | | ✓ |
| QNLI (Wang et al., 2018) | accuracy | 52.65 | **51.35** | 50.94 | 50.28 | 49.99 | 50.05 | ✓ | |
| MMLU (Hendrycks et al., 2021) | accuracy | **50.54** | 38.30 | 47.86 | 37.59 | 46.88 | 49.46 | ✓ | |
| RACE (Lai et al., 2017) | accuracy | 37.22 | 39.23 | **39.62** | 37.42 | 38.28 | **39.62** | ✓ | |
| MathQA (Amini et al., 2019) | accuracy | 29.92 | 27.67 | 29.85 | 26.93 | 30.62 | **31.02** | | ✓ |
| SciQ (Welbl et al., 2017) | accuracy | 94.60 | 91.20 | 94.40 | 91.00 | 94.20 | **94.70** | ✓ | |
| PROST (Aroca-Ouellette et al., 2021) | accuracy | 28.24 | 25.80 | 28.99 | 26.80 | 27.84 | **29.16** | ✓ | |
| BoolQ (Clark et al., 2019) | accuracy | 74.50 | 71.96 | 76.64 | 68.59 | 75.96 | **77.77** | ✓ | |
| COPA (Roemmele et al., 2011) | accuracy | **89.00** | 80.00 | 85.00 | 75.00 | 86.00 | **89.00** | | ✓ |
| LogiQA (Liu et al., 2021) | accuracy | **23.35** | 23.04 | 23.04 | 22.73 | 20.89 | 22.73 | ✓ | |
| COQA (Reddy et al., 2019) | exact match score | **56.42** | 50.90 | 56.37 | 47.63 | 53.23 | 55.35 | ✓ | |
| COQA (Reddy et al., 2019) | F1 score | **71.20** | 66.17 | 70.99 | 63.28 | 68.85 | 70.84 | ✓ | |

For Qwen-MoE, we provide the results in Table 4, where the raw *off-the-shelf* model performs best on 5 benchmarks. Apart from these tasks, router-based pruning outperforms similarity-based pruning on 9 tasks, while similarity-based pruning outperforms router-based pruning on 11 tasks.

*Table 4.* **Full Evaluation Results for Collaboration Pruning with Qwen-MoE.**

| Method | Metric | Raw | Router-based Prune, 1 GPU | Router-based Prune, 2 GPUs | Sim-based Prune, 1 GPU | Sim-based Prune, 2 GPUs | Standard SFT | Router-based Better | Sim-based Better |
|---|---|---|---|---|---|---|---|---|---|
| WSC (Levesque et al., 2012) | accuracy | 82.05 | 79.49 | 81.68 | 78.39 | **82.78** | 80.59 | | ✓ |
| Winogrande (Sakaguchi et al., 2021) | accuracy | 68.43 | 68.11 | 69.06 | 66.06 | **70.24** | 70.01 | | ✓ |
| ASDiv (Miao et al., 2020) | accuracy | 4.38 | **16.36** | 12.28 | 12.49 | 10.63 | 11.28 | ✓ | |
| OpenBookQA (Mihaylov et al., 2018) | accuracy | 30.40 | 27.80 | 29.40 | 28.60 | **32.20** | 29.60 | | ✓ |
| PIQA (Bisk et al., 2020) | accuracy | 79.71 | 78.62 | 80.25 | 78.51 | 79.87 | **80.36** | ✓ | |
| HellaSwag (Zellers et al., 2019) | accuracy | **57.95** | 55.29 | 57.19 | 54.73 | 57.74 | 57.42 | | ✓ |
| SST-2 (Socher et al., 2013) | accuracy | 68.46 | 78.21 | 77.41 | 73.17 | **87.04** | 82.22 | | ✓ |
| MultiNLI (Williams et al., 2018) | accuracy | 49.77 | 49.83 | 51.88 | 46.41 | **54.40** | 52.50 | | ✓ |
| QASPER (Dasigi et al., 2021) | F1 score | 90.81 | **98.78** | 97.03 | 98.04 | 86.65 | 97.54 | ✓ | |
| MRPC (Dolan & Brockett, 2005) | accuracy | 76.47 | 71.57 | 78.19 | 76.72 | 75.49 | **78.92** | ✓ | |
| MRPC (Dolan & Brockett, 2005) | F1 score | **85.62** | 82.48 | 85.19 | 83.76 | 84.52 | 83.10 | ✓ | |
| MultiRC (Khashabi et al., 2018) | accuracy | 37.50 | **43.15** | 39.62 | 38.72 | 40.14 | 36.67 | ✓ | |
| WNLI (Wang et al., 2018) | accuracy | **57.75** | 43.66 | 54.93 | 56.34 | 56.34 | 54.93 | | ✓ |
| RTE (Bentivogli et al., 2009) | accuracy | 68.23 | 69.68 | 69.68 | 65.34 | **75.45** | 71.12 | | ✓ |
| QNLI (Wang et al., 2018) | accuracy | **57.44** | 53.25 | 56.05 | 52.00 | 51.18 | 54.99 | ✓ | |
| MMLU (Hendrycks et al., 2021) | accuracy | **60.86** | 56.85 | 60.13 | 56.23 | 57.25 | 59.99 | ✓ | |
| RACE (Lai et al., 2017) | accuracy | 39.43 | 40.67 | 41.34 | 39.43 | 41.05 | **41.72** | ✓ | |
| MathQA (Amini et al., 2019) | accuracy | 36.15 | 36.48 | 38.93 | 37.09 | **41.31** | 39.26 | | ✓ |
| SciQ (Welbl et al., 2017) | accuracy | 94.40 | 95.40 | 95.60 | **95.70** | 95.50 | 95.20 | | ✓ |
| PROST (Aroca-Ouellette et al., 2021) | accuracy | 30.50 | 30.01 | 31.22 | 30.08 | **32.47** | 31.41 | | ✓ |
| BoolQ (Clark et al., 2019) | accuracy | 79.57 | 78.38 | 79.88 | 77.22 | **81.41** | 80.40 | | ✓ |
| COPA (Roemmele et al., 2011) | accuracy | 84.00 | 81.00 | **86.00** | 79.00 | 83.00 | 83.00 | ✓ | |
| LogiQA (Liu et al., 2021) | accuracy | 30.41 | 30.26 | 31.03 | 31.80 | **31.95** | 29.03 | | ✓ |
| COQA (Roemmele et al., 2011) | exact match score | 64.40 | 65.93 | **66.77** | 66.28 | 64.48 | 66.23 | ✓ | |
| COQA (Roemmele et al., 2011) | F1 score | 78.60 | 78.15 | **80.04** | 79.30 | 77.53 | 79.48 | ✓ | |

For DeepSeek-MoE, we provide the results in Table 5, where the raw *off-the-shelf* model performs best on 7 benchmarks. Apart from these tasks, router-based pruning outperforms similarity-based pruning on 11 benchmarks, while similarity-based pruning outperforms router-based pruning on 7 benchmarks.

*Table 5.* **Full Evaluation Results for Collaboration Pruning with DeepSeek-MoE.**

| Method | Metric | Raw | Router-based Prune, 1 GPU | Router-based Prune, 2 GPUs | Sim-based Prune, 1 GPU | Sim-based Prune, 2 GPUs | Standard SFT | Router-based Better | Sim-based Better |
|---|---|---|---|---|---|---|---|---|---|
| WSC (Levesque et al., 2012) | accuracy | **84.98** | 78.75 | 83.88 | 82.05 | 84.62 | 84.62 | | ✓ |
| Winogrande (Sakaguchi et al., 2021) | accuracy | 70.48 | 66.30 | 70.40 | 66.22 | 70.72 | **71.19** | | ✓ |
| ASDiv (Miao et al., 2020) | accuracy | 0.91 | 4.77 | 2.82 | 1.65 | 3.08 | **4.95** | ✓ | |
| OpenBookQA (Mihaylov et al., 2018) | accuracy | 32.20 | 29.60 | 33.40 | 29.20 | **34.20** | **34.20** | | ✓ |
| PIQA (Bisk et al., 2020) | accuracy | 78.73 | 77.15 | **79.92** | 77.26 | 79.76 | 79.11 | ✓ | |
| HellaSwag (Zellers et al., 2019) | accuracy | 58.06 | 54.05 | 58.36 | 53.71 | 58.31 | **58.49** | ✓ | |
| SST-2 (Socher et al., 2013) | accuracy | 64.68 | 65.71 | 76.72 | 59.40 | 73.05 | **78.33** | ✓ | |
| MultiNLI (Williams et al., 2018) | accuracy | 42.30 | 36.77 | **45.65** | 38.01 | 44.42 | 41.55 | ✓ | |
| QASPER (Dasigi et al., 2021) | F1 score | 93.06 | 90.24 | 95.74 | **98.04** | 97.03 | 91.67 | | ✓ |
| MRPC (Dolan & Brockett, 2005) | accuracy | **68.63** | 68.14 | 68.38 | 67.65 | 68.38 | 68.38 | ✓ | |
| MRPC (Dolan & Brockett, 2005) | F1 score | **81.29** | 80.71 | 81.17 | 80.70 | 81.22 | 81.22 | | ✓ |
| MultiRC (Khashabi et al., 2018) | accuracy | **57.03** | 54.68 | 56.72 | 56.60 | 56.68 | 57.01 | ✓ | |
| WNLI (Wang et al., 2018) | accuracy | 50.70 | 43.66 | 45.07 | 45.07 | **52.11** | 47.89 | | ✓ |
| RTE (Bentivogli et al., 2009) | accuracy | 62.82 | 61.01 | 62.09 | 61.01 | **65.34** | 60.29 | | ✓ |
| QNLI (Wang et al., 2018) | accuracy | 49.50 | **53.12** | 50.14 | 50.17 | 50.80 | 49.83 | ✓ | |
| MMLU (Hendrycks et al., 2021) | accuracy | 37.95 | 36.16 | **42.43** | 34.69 | 41.91 | 38.66 | ✓ | |
| RACE (Lai et al., 2017) | accuracy | 38.85 | 38.56 | 39.81 | 38.47 | 39.62 | **40.10** | ✓ | |
| MathQA (Amini et al., 2019) | accuracy | 31.19 | 29.21 | 33.50 | 30.25 | 33.63 | **33.77** | | ✓ |
| SciQ (Welbl et al., 2017) | accuracy | 92.80 | 93.30 | 93.50 | **94.70** | 93.50 | 93.40 | ✓ | |
| PROST (Aroca-Ouellette et al., 2021) | accuracy | 28.72 | 28.26 | **29.60** | 28.91 | 28.79 | 28.64 | ✓ | |
| BoolQ (Clark et al., 2019) | accuracy | 72.39 | 68.69 | 68.87 | **73.39** | 70.15 | 71.93 | | ✓ |
| COPA (Roemmele et al., 2011) | accuracy | **90.00** | 84.00 | 89.00 | 86.00 | 87.00 | 88.00 | ✓ | |
| LogiQA (Liu et al., 2021) | accuracy | 25.35 | 24.42 | **25.96** | 25.65 | 25.65 | **25.96** | ✓ | |
| COQA (Reddy et al., 2019) | exact match score | **64.15** | 64.02 | 63.83 | 62.25 | 62.80 | 63.17 | ✓ | |
| COQA (Reddy et al., 2019) | F1 score | **77.79** | 77.38 | 77.33 | 75.46 | 77.05 | 76.72 | ✓ | |

