# OpenReview forum: "Occult: Optimizing Collaborative Communications across Experts for Accelerated Parallel MoE Training and Inference"
_ICML.cc/2025/Conference — ICML 2025 poster_

### Official Review · Reviewer_NL62 · 2025-03-11

**Overall Recommendation:** 3

**Summary:**

All-to-all communication is a major bottleneck in training and inference for mixture-of-experts (MoE) large language models. While existing MoE kernels have improved computational efficiency, all-to-all communication remains a bottleneck. The authors propose Occult, which aims to (1) reduce redundant communication and (2) encourage in-device collaboration via router update to further optimize latency.

**Claims And Evidence:**

Most claims are well-supported by evidence or align with well-established knowledge in the field. However, some claims lack sufficient clarity or appropriate experimental validation:

- The claim that the proposed algorithm benefits training is not adequately evaluated. (See below)
- The paper does not provide the corresponding code, making it difficult to verify reproducibility.

**Essential References Not Discussed:**

I'm not aware of any essential references that were not discussed.

**Experimental Designs Or Analyses:**

The experimental design appears insufficient to fully support the paper’s claims:

- The stated goal is to reduce all-to-all latency, which is especially problematic in multi-node multi-gpu setups. However, the evaluation only uses four GPUs connected via PCIe, which has limited bandwidth. This setup leads to a high number of experts per GPU and does not reflect realistic multi-node scenarios. While the resource constraints are understandable, without larger-scale results, the claim about latency reduction remains unconvincing. The author should provide more results, either from experiments or projections, to support this claim. Without the new evidence, the proposed method might be more applicable for **inference**, and the paper should consider revising its scope to emphasize this contribution.

For section 5.2:
- The comparison is misleading: fine-tuning naturally improves model performance, so a fine-tuned model outperforming a baseline is expected. If the goal is to show that "expanding to two devices achieves comparable or superior quality to standard training," the baseline should also be fine-tuned for a fair comparison.


**I hope the author could address these concerns, and I'm more than willing to update my evaluation if my concerns are addressed.**

**Methods And Evaluation Criteria:**

The proposed method and evaluation criteria are well-aligned with the problem at hand.

**Other Comments Or Suggestions:**

Line 218 right column: continuously -> contiguously
Line 383: device amount -> device count

**Other Strengths And Weaknesses:**

While I think the paper do provide some interesting insight on MoE, the writing of this paper has clarity issues, and should be significantly improved. Sections 3 and 4 are difficult to follow and should be rewritten for clarity. Here are some examples:

- Figure 2
    - This figure is very confusing. Readers unfamiliar with MoE literature will struggle to interpret it. The term device D-0/1 appears to refer to physical devices, but it actually denotes expert assignments of token. The figure needs revision.

- Communication Complexity in Section 3.2
  - The term "communication complexity" is misleading. A more accurate term would be "communication volume" to reflect what is actually being measured, but it seems to be different from the definition used in Figure 2.

- Figure 4
  -There is no in-text reference to Figure 4. The figure itself is unclear and needs better explanation.

**Questions For Authors:**

Why is Hugging Face’s decoding latency lower than the proposed method when batch size is small?

**Relation To Broader Scientific Literature:**

The proposed optimization reduces all-to-all latency, a well-known bottleneck in MoE-based LLMs. The work contributes to this active research area.

**Theoretical Claims:**

Equation (5) states that communication complexity is lower-bounded by the number of devices needed to fill k and upper-bounded by the minimum of k and the number of devices. While this holds in terms of send/receive operations, this definition differs from the one used in Figure 2 and Section 3.1, creating inconsistency.

---

> ### Author Rebuttal · Authors · 2025-04-01
>
> We thank reviewer NL62 for the dedicated and professional comments. To address your concerns, we provide detailed pointwise responses below:
>
> **[Claims and Evidence]**
>
> We provide code at https://anonymous.4open.science/r/Occult-D802.
>
> **[Theoretical claim 1: Communication complexity in Fig. 2]**
>
> Thank you for the careful review. We admit there's a typo in Fig. 2(c): the blue token routed to Expert 1 and the yellow one routed to Expert 2 should be exchanged due to our intentional modification of routing choices to ensure tokens only activate experts on the same device, thereby reducing all-to-all communication volume.
>
> To clarify the communication complexity in Fig. 2:
> - Fig. 2(a): Each token is repeated twice, yielding $C_{\mathcal{T}}$ = 2
> - Fig. 2(b): Red and green tokens are repeated once while yellow and blue tokens are twice, yielding $C_{\mathcal{T}}$ = 1.5
> - Fig. 2(c): Each token is repeated once, yielding $C_{\mathcal{T}}$ = 1
>
> The confusion may stem from insufficient emphasis on top-k value. This example uses 2 devices, 4 tokens, 4 experts, and top-2 routing. According to our derivation in Section 3.2, communication complexity bounds are $1\leq C_{\mathcal{T}}\leq 2$, with all 3 cases in Fig. 2 falling in this interval. We will revise Fig. 2 and clearly explain this modification in Fig. 2(c) to make it more comprehensible.
>
> **[Experiments & Analysis 1: Multi-node Training]**
>
> Our research is targeted at communication efficiency, so it inherently performs better than conventional expert parallelism on multi-node training, where inter-node communication is the main bottleneck. We further examine 8-way (1 node ) and 16-way ( 2 nodes ) expert-parallelized training for DeepSeek-MoE, where latency and evaluation comparison are also visualized at https://anonymous.4open.science/r/Occult-D802.
>
> **[Experiments & Analysis 2: Fine-tuning baseline for fair comparison]**
>
> In Sec. 5.2 and Fig. 7, we provide the evaluation results for
> - Original model (no tuning), shown in yellow dashed line
> - Tuned model with pruning, shown in brown dots ( similarity-based pruning ) and green stars ( router-based pruning ).
> - Standard tuning, shown in pink diamond.
>
> In the caption text of Fig. 7, we indicated that pruning within 4 devices is equivalent to standard tuning since we use 4 devices for distributed training. Therefore, the effect of pruning can be derived by comparing the brown dots & green stars with pink diamonds. The yellow line serves as a reference.
>
> **[Strengths and weaknesses 1: Fig. 2]**
>
> Thank you for the feedback. We will improve the caption to clarify the meaning of $D_i$, $D_i^j$, and $E_i$, and add explicit in-text references to better guide readers through this illustration of different communication strategies.
>
> **[Strengths and weaknesses 2: Communication complexity]**
>
> We use communication complexity to indicate the ratio of all-to-all communication volume to the number of tokens, approximated by the average token replication count. This approach addresses two key challenges:
> - During all-to-all communication, a token replica may either be retained locally or transmitted to other devices based on its routing choice, posing uncertainty in precisely measuring inter-device communication volume across different tasks.
> - To establish a more stable metric, we use average token replication times, which is also the least upper bound for the ratio of inter-device communication volume to token count, minimizing the impact of dynamic routing uncertainties.
>
> We appreciate you highlighting this concern and will expand these explanations in Section 3.2 to provide better clarity.
>
> **[Strengths and weaknesses 3: Figure 4]**
>
> We apologize for the missing in-text reference to Fig. 4. We will add detailed descriptions, including:
> - $\texttt{SFD}$ tokens serve as the all-to-all content, constructed from $\texttt{ORI}$ tokens based on $BRIM_0$
> - $\texttt{FFN}$ utilizes $BRIM_1$ as an auxiliary input to guide the token layout of the output tensor
> - Intermediate tokens are organized densely in the $\texttt{EPD}$ state
>
> I hope they can help readers better understand the token state transitions and the role of $BRIM$s in our framework.
>
> **[Other comments]**
>
> Thanks for pointing out. We will fix these typos carefully in our revised draft.
>
> **[Question: Huggingface Latency]**
>
> Prefilling latency is the primary bottleneck when generating a small number of tokens. Hugging Face’s native API employs pipeline parallelism (PP), which outperforms expert parallelism (EP) under the following conditions:
> - Limited GPU amount ( our experiments use only 4 GPUs )
> - Deep model architectures ( the three evaluated models contain 16, 24, and 28 layers, respectively )
>
> Thereby, the communication volume of EP in the prefilling stage is much larger than PP. However, the decoding speed of EP is much faster, profit by the efficient MoE computation.
>
> Thank you again for your comments. We will incorporate these refinements in our revised draft.

---

> > ### Comment · Reviewer_NL62 · 2025-04-04
> >
> > I appreciate the authors' effort to open-source their code as well as providing additional evaluation results. I suppose the number of GPUs mentioned in the README file "Occult (Prune, 1 GPU)" actually refers to the pruning count $N_d$? If that's the case, perhaps the authors may want to provide additional explanation in its caption, like they did in Figure 8. I have increased my rating to reflect my latest evaluation of this paper. I hope the authors can further improve the readability of this manuscript should it be accepted.

---

> > > ### Author Response · Authors · 2025-04-05
> > >
> > > We sincerely appreciate reviewer NL62 for increasing the rating. You're right, the number of GPUs in caption "Occult (Prune, 1 GPU)" exactly refer to the pruning count $N_d$. In this case, we prune the expert collaborations for each token so that an individual token only activates experts within 1 GPU. We have updated the code repository and replaced the table with 2 figures, similar to Figure 11 in our manuscript. We also provide detailed explanations in the captions of figures and tables in this README file. In our revised version, we will expand Figure 11 into 3 sub-figures, and enrich its caption by providing these additional explanations, just like Figure 8. Thank you for your detailed suggestions.

---

### Official Review · Reviewer_1GwW · 2025-03-13

**Overall Recommendation:** 4

**Summary:**

The paper introduces Occult, an algorithm-system co-design approach to optimize collaborative communication in MoE models for large-scale training and inference. The key idea is to reduce inter-device communication costs by maximizing intra-device expert collaboration, using expert placement rescheduling and collaboration pruning strategies. The paper shows that Occult achieves over 50% speedup across various MoE-based LLMs while maintaining or even improving model quality. The authors provide theoretical justifications, empirical validation, and extensive benchmarking against state-of-the-art frameworks like DeepSpeed, Tutel, and MegaBlocks.

**Claims And Evidence:**

The paper claims that optimizing collaborative communication via expert placement rescheduling and collaboration pruning can significantly reduce all-to-all communication costs in MoE models, leading to faster training and inference with minimal quality degradation.

**Support:**

(a) Theoretical derivations: The authors introduce a collaboration graph-based formulation to quantify inter- and intra-collaboration.

(b) Algorithm side: The expert placement rescheduling algorithm is validated through profiling experiments showing up to 20% reduction in communication budget.

(c) Empirical evaluation: Occult achieves up to 8.66× speedup for inference and up to 10× faster training over baseline MoE frameworks.

**Potential concerns:**

(a) The impact of aggressive collaboration pruning on model quality could be better explored for different task types.

(b) No direct discussion on scalability beyond 4 GPUs—it would be useful to see how Occult performs on larger clusters and models.

**Essential References Not Discussed:**

There are no specific references currently on my mind that have not been discussed.

**Experimental Designs Or Analyses:**

Experimental setup is generally robust, with:
(a) Multiple MoE architectures (OLMoE, DeepSeek-MoE, Qwen-MoE) (2) Comparison against strong baselines (DeepSpeed, Tutel, MegaBlocks). (3) Latency benchmarks covering different workloads (training, inference, decoding).

Limitations or improvements:
(a) Limited scalability analysis beyond 4 GPUs. (b) The selection of expert placement rescheduling is heuristic—would an end-to-end learned approach perform better?

**Methods And Evaluation Criteria:**

The proposed methods make sense for optimizing communication bottlenecks in MoE training and inference. Evaluation criteria are appropriate, leveraging: (a) Latency benchmarks for training and inference (Figures 9–11). (b) Accuracy and task performance metrics across multiple NLP benchmarks. The concern is that the paper does not explicitly evaluate multi-node scaling performance, and the evaluated model is relatively small scale.

**Other Comments Or Suggestions:**

1. What are required to be done to scale up Occult when moving beyond single-node multi-GPU setups?
2. Instead of heuristics, could an RL-based expert assignment policy be more effective?

**Other Strengths And Weaknesses:**

Pros:
1. The paper addresses a key limitation in MoE scalability. Communication is a well-known challenge in distributed MoE training, and Occult provides a practical, well-validated optimization.
2. The paper presents strong empirical results, a 50%+ speedup in multiple MoE workloads is a compelling result.
3. A lightweight, heuristic expert placement rescheduling was proposed that provides significant efficiency gains.

Cons:
1. Limited discussion on multi-node scalability. The paper mainly explores single-node, multi-GPU setups, leaving open questions about large-cluster scaling.
2. Trade-offs in collaboration pruning are not well studied. More ablation studies would help clarify when pruning impacts accuracy.

**Questions For Authors:**

Please check "Other Comments Or Suggestions"

**Relation To Broader Scientific Literature:**

The paper is well-grounded in prior work, citing relevant MoE frameworks (DeepSpeed-MoE, Tutel, MegaBlocks). The connection to general parallel computing strategies could be stronger—Occult shares similarities with load balancing and scheduling techniques in distributed systems.

**Theoretical Claims:**

The paper does not introduce new mathematical proofs, but it provides well-founded theoretical insights, including quantification of communication cost and its relationship with intra- and inter-collaboration; and the optimization bounds for communication overhead in MoE expert parallelism.

---

> ### Author Rebuttal · Authors · 2025-04-01
>
> We thank Reviewer 1GwW for recognizing that "the paper addresses a key limitation in MoE scalability", “experiments generally robust”, and that "Occult provides a practical, well-validated optimization." To address your questions, we provide pointwise responses below.
>
> **[Potential concerns 1: Different task types]**
>
> We've evaluated the performance on 23 benchmarks with different tuning strategies in Tab. 3,4,5 on pages 16, 17, and 18, covering natural language understanding, commonsense reasoning, and math. Besides, we conduct additional experiments on HumanEval (coding) and GSM8K (math) datasets for DeepSeek-MoE. The results below consistently demonstrate the effectiveness of our efficient pruning methods:
>
> HumanEval:
>
> |Method|No Tune|Prune, 1 GPU|Prune, 2 GPUs|No Prune|
> |-|-|-|-|-|
> |Router-based|26.83|17.68|**27.44**|22.56|
> |Sim-based|26.83|17.07|26.22|22.56|
>
> GSM8K ( flexible-extract ):
>
> |Method|No Tune|Prune, 1 GPU|Prune, 2 GPUs|No Prune|
> |-|-|-|-|-|
> |Router-based|16.91|7.28|15.31|**17.97**|
> |Sim-based|16.91|11.22|16.00|**17.97**|
>
> **[Potential concerns 2: Scalability]**
>
> Our occult is more feasible for modern fine-grained MoE-LLMs such as DeepSeek-MoE. Sadly, models with a scale larger than 16B are out of our capability to train ( such as DeepSeek-V2 ). We conduct additional experiments on 8 GPUs and 2 x 8 GPUs (two nodes) to demonstrate the scalability of our method, using DeepSeek-MoE with 8- and 16-way expert parallelism, taking batch size 32 per GPU:
>
> Avg training latency per step (s):
>
> |Setting|Occult ( 1 GPU )|Occult ( 2 GPUs )|Occult ( 3 GPUs )|Occult ( 4 GPUs )|MegaBlocks|
> |-|-|-|-|-|-|
> |8 GPUs|8.50|9.31|10.95|11.92|16.56|
> |16 GPUs||9.55|10.25|10.93|14.97|
>
> Occult acceleration can be more apparent on a better-grouped expert placement, thereby Occult can better improve training efficiency on 8-way expert parallelism.
>
> We also evaluate 8- and 16-way EP on MMLU and MathQA:
>
> 8-way EP:
>
> |Task|Strategy|No Tune|Prune within 1 GPU|Prune within 2 GPUs|Prune within 3 GPUs|Prune within 4 GPUs|Prune within 5 GPUs|No Prune|
> |-|-|-|-|-|-|-|-|-|
> |MMLU|Router-based|37.95|35.04|40.41|41.34|41.43|41.19|38.66|
> |MMLU|Sim-based|37.95|33.68|39.80|**41.74**|41.40|41.48|38.66|
> ||
> |MathQA|Router-based|31.19|32.93|35.08|34.97|35.95|**36.08**|33.77|
> |MathQA|Sim-based|31.19|33.17|34.94|35.51|35.24|35.61|33.77|
>
> 16-way EP:
>
> |Task|Strategy|No Tune|Prune within 2 GPUs|Prune within 3 GPUs|Prune within 4 GPUs|Prune within 5 GPUs|No Prune|
> |-|-|-|-|-|-|-|-|
> |MMLU|Router-based|37.95|39.69|40.37|41.23|**41.62**|38.66|
> |MMLU|Sim-based|37.95|39.23|40.25|41.31|41.61|38.66|
> ||
> |MathQA|Router-based|31.19|35.61|35.14|35.21|**35.78**|33.77|
> |MathQA|Sim-based|31.19|34.84|35.21|35.68|35.71|33.77|
>
> **[Strengths and Weaknesses 2: Collaboration pruning trade-offs]**
>
> We've analyzed the impact of different pruning settings on both performance and efficiency in Fig. 7, 8, 10, and 11. A more comprehensive accuracy analysis is provided in Tab. 3,4,5 on page 16.17.18.  Additionally, we conduct an additional ablation study to analyze the pruning impact on efficiency with DeepSeek, using 8-way EP:
>
> ||Megablocks|Occult (w/o Prune)|Occult (Prune, 4 GPUs)|Occult (Prune, 3 GPUs)|Occult (Prune, 2 GPUs)|Occult (Prune, 1 GPUs)|
> |-|-|-|-|-|-|-|
> |Memory (GB)|43.12|40.68|36.93|34.46|32.18|30.73|
> |Avg Latency per Step (s)|8.36|6.43|6.02|5.54|4.44|3.93|
>
> **[Comments 1: To multi-node setting]**
>
> Thanks for the practical question. To scale up our method to the multi-node settings, we further conduct extra experiments on multi-node settings, as shown in [Potential concerns 2: Scalability], and demonstrate the scalability of our method. To scale it to a very large cluster, a combined parallel strategy for the MoE layer is required to adapt to the hardware resources, including:
> - Data parallelism is required to repeat the basic unit of expert parallelism (e.g., 16 GPUs in 16-way EP ), aiming at avoiding heavy communication overhead across nodes ( e.g., repeating the parameter of each MoE layer for 4 times in a 64-GPU cluster )
> - Expert parallelism can be organized across different nodes, placing the grouped experts in Occult on the same node
> - Tensor parallelism can be adopted in intra-node GPUs for very large expert since the intra-node bandwidth is usually abundant
>
> **[Comments 2: RL-based expert assignment]**
>
> End-to-end rescheduling may be impracticable, as dynamically tuning expert placement requires huge additional bandwidth, although it can be asynchronous.
>
> While our current heuristic approach yields strong results, we believe our expert assignment algorithms can be further improved for enhanced performance. RL-based methods can learn the collaborative communication patterns from the profiling dataset, which can benefit the expert assignment task and improve generalization. We have not tried similar methods yet, but it is promising for our future research.
>
> Thank you again for your questions. We will incorporate the refinements in our revised draft.

---

### Official Review · Reviewer_geV9 · 2025-03-14

**Overall Recommendation:** 4

**Summary:**

In this paper, the author proposes Occult, an MoE training and inference framework designed to reduce communication costs by effectively managing intra- and inter-collaboration among experts. The evaluation results demonstrate that the proposed method achieves significant speedup compared to the state-of-the-art MoE framework.

**Claims And Evidence:**

Overall, the evidence is sufficient to support the claims.

**Essential References Not Discussed:**

The references provided are appropriate and sufficient.

**Experimental Designs Or Analyses:**

1. Model configurations, such as the number of experts, are critical but not provided.

2. It appears that different pruning strategies significantly affect performance (e.g., similarity-based and router-based). However, there is no empirical or theoretical analysis to determine which strategy should be utilized under specific scenarios.

3. Regarding Figures 9 and 10, presenting throughput (tokens/s), TTFT, and TPOT under varying batch sizes and sequence lengths may be more common, useful, and informative for evaluating performance.

**Methods And Evaluation Criteria:**

Overall, the baselines and datasets are appropriate.

**Other Comments Or Suggestions:**

The notation "0.66x faster" and "0.55x speedup" seems to indicate a slowdown rather than a speedup. Maybe it should be 1.66x faster and 1.55x speedup.

**Other Strengths And Weaknesses:**

1. There is no discussion on the pruning cost or the impact of bandwidth.

2. The performance may heavily depend on the number of GPUs. Since the evaluations are conducted with 4 GPUs, there is a greater opportunity to increase intra-collaboration. However, for larger models trained with more GPUs, such as 1,000 GPUs, it will be more challenging to place experts on the same device, or model performance may be compromised to achieve this.

**Questions For Authors:**

1. What is the cost of pruning? Does it require more time to converge?

2. What is the impact of bandwidth? It seems that the improvement is more significant in a low-bandwidth system. What would the improvement be in a high-bandwidth system?

3. DeepSeek recently released DeepEP, a communication library tailored for MoE and EP. Is your optimization orthogonal to DeepEP? Would your method benefit from DeepEP or achieve even better performance if integrated with it? I believe this is a parallel work, so the discussion is not required but welcome.

**Relation To Broader Scientific Literature:**

This work enhances the training and inference efficiency of MoE models, which is crucial for future LLM deployment.

**Theoretical Claims:**

Overall, the theoretical claims are correct.

---

> ### Author Rebuttal · Authors · 2025-04-01
>
> We sincerely thank reviewer geV9 for recognizing that our approach "enhances the training and inference efficiency of MoE models, which is crucial for future LLM deployment."  To address your questions, we provide pointwise responses below.
>
> **[Experiments & Analysis 1: Model configuration]**
>
> Thanks for the constructive feedback. We've added a supplementary table detailing model configurations:
>
> |Model|# Params|# Experts|Top-K|# MoE layers|Hidden size|Expert intermediate size|
> |-|-|-|-|-|-|-|
> |OLMoE|6.92B|64|8|16|2048|1024|
> |Qwen1.5-MoE|14.3B|60|4|24|2048|1408|
> |DeepSeek-MoE|16.4B|64|6|27|2048|1408|
>
> **[Experiments & Analysis 2: Pruning strategies]**
>
> The performance differences between router- and similarity-based pruning arise from how they handle expert replacement:
> - Router-based pruning uses only routing scores, which may not involve expert attribute
> - Similarity-based pruning considers inter-expert similarity, obtained from a profiling dataset
>
> As shown in Figure 7, similarity-based pruning achieves comparable or better performance than router-based approaches in most cases of models and tasks. For example, in the OLMoE results on RTE, similarity-based pruning (shown as "Pruning (Similarity-based)") achieves approximately 3% higher accuracy than router-based pruning when restricting collaboration to 1 GPU.
>
> Similarity-based pruning can utilize prior knowledge from profiling datasets, which may advance the downstream tasks under similar domain distribution. However, it would expire on out-of-distribution tasks, where router-based pruning may perform better.
>
> **[Experiments & Analysis 3: throughput, TTFT, and TPOT]**
>
> Following your valuable suggestion, we will adopt “throughput (tokens/s), Time To First Token (TTFT), and Time Per Output Token (TPOT)” in Fig. 9 & 10 under varying batch sizes and sequence lengths in the formats below:
>
> Prefilling:
>
> |Model|Method|# Tokens|Throughput|TTFT (s)|
> |-|-|-|-|-|
> |OLMoE|Occult (Prune, 2 GPUs)|16384|1777.01|9.22|
>
> Decoding:
>
> |Model|Method|# Generated Tokens|Throughput|TPOT (ms)|
> |-|-|-|-|-|
> |OLMoE|Occult (Prune, 2 GPUs)|512|682.67|1.46|
>
> **[Strengths And Weaknesses 1: Cost of pruning]**
>
> Pruning does not require more time to converge because
> - Results in Fig. 7 experiments are all reported with 1 epoch training on the Alpaca dataset, on some tasks it can even perform better than standard SFT.
> - The average latency per training step can be greatly reduced with Occult.
>
> We report the latency and memory cost of pruning for DeepSeek-MoE here, with batch size 16 and 8 GPUs:
>
> ||Megablocks|Occult (w/o Prune)|Occult (Prune, 4 GPUs)|Occult (Prune, 3 GPUs)|Occult (Prune, 2 GPUs)|Occult (Prune, 1 GPUs)|
> |-|-|-|-|-|-|-|
> |Memory (GB)|43.12|40.68|36.93|34.46|32.18|30.73|
> |Avg Latency per Step (s)|8.36|6.43|6.02|5.54|4.44|3.93|
>
> Our prune algorithms can accelerate clock-time latency and reduce memory costs for training.
>
> **[Strengths And Weaknesses 2: Bandwidth impact]**
>
> We conducted additional experiments with varying interconnect bandwidths on different machines. The acceleration of Occult is more significant with lower bandwidth. We use DeepSeek-MoE with 8-way EP and batch size 32:
>
> |Bandwidth|Speedup (Occult vs. Megablocks, w/o Prune)|Speedup (Occult vs. Megablocks, Prune, 2 GPUs)|Speedup (Occult vs. Megablocks, Prune, 1 GPU)|
> |-|-|-|-|
> |18GB/s|1.37|1.78|1.95|
> |46GB/s|1.12|1.43|1.76|
>
> **[Strengths And Weaknesses 3: Scalability]**
>
> Our approach can remain effective at a very large scale with a hierarchical strategy:
> - Intra-node optimization: Within each node (typically 8 GPUs connected via NVLink), we can place multiple experts on the same node to maintain high intra-collaboration within the node boundary, where communication is fast.
> - Inter-node optimization: Cross nodes, expert placement rescheduling & collaboration pruning can be applied to minimize cross-node communication, which is typically the latency bottleneck.
>
> This scaling profile can be beneficial since inter-node communication is often 2-10× slower than intra-node communication, making the reduction of cross-node traffic especially valuable. In large-scale deployments, Occult's benefits should increase rather than diminish as communication overhead becomes more dominant.
>
> **[Comments: Notation issue]**
>
> Thanks for the constructive suggestion on “faster” and “speedup”, we will fix it in the revised paper.
>
> **[Question 3: Integration with DeepEP]**
>
> DeepEP mainly contains 2 stages for hardware-tailored MoE communication:
> - Inter-node all-to-all: sending tokens to the corresponding GPU on the target node
> - Intra-node all-to-all: sending tokens to the target GPUs containing the target expert
>
> In our understanding, Occult is orthogonal to DeepEP since it optimizes all-to-all communication volume along the dimension of top-k aggregation, which can be combined with DeepEP for enhanced efficiency.
>
> Thank you again for your questions. We will include these additional experiments and analyses in our revised draft.

---

> > ### Comment · Reviewer_geV9 · 2025-04-06
> >
> > Thank you for your responses. I have no further questions and will keep my original score.

---

### Official Review · Reviewer_tR7y · 2025-03-14

**Overall Recommendation:** 3

**Summary:**

This paper introduces Occult, an algorithm-system co-design approach aimed at reducing the communication overhead of Mixture-of-Experts (MoE) large language models (LLMs). Specifically, the authors first propose BRIM, a data structure designed to support fundamental MoE operations efficiently. Next, they optimize expert placement based on calibration data to minimize inter-device communication. Finally, by replacing experts on remote devices with similar experts on the N_d devices, they further reduce collaborative communication overhead. Experimental results demonstrate that the proposed method significantly improves efficiency across various scenarios, including prefilling, decoding, and training, compared to existing frameworks.

**Claims And Evidence:**

Yes

**Essential References Not Discussed:**

No

**Experimental Designs Or Analyses:**

Issues:

- The experiments can be further improved. It is better for the authors to conduct experiments on larger MoE and MoE with fewer number of experts (e.g., Mixtral), which will demonstrate the generalization and scaling ability of the method.

- According to Table 3 and the introduction section, the proposed method can achieve higher performance than the original model. However, the authors only provide explanation on why two-device pruning is better than single-device pruning in Sec 5.2. More explanation is needed regarding the former one.

**Methods And Evaluation Criteria:**

Yes

**Other Comments Or Suggestions:**

No

**Other Strengths And Weaknesses:**

**Strengths**
1. The motivation of the paper is clear. Collaborative communication is a major overhead in MoE computation, and optimizing it through co-design of hardware and algorithms is a reasonable approach.

2. Experiments show that the proposed method achieves faster speed and higher performance compared to baseline methods.

**Weakness**
Please refer to "Experimental Designs Or Analyses"

**Questions For Authors:**

The authors provides a new framework for all-to-all communication in Sec 4.1. What's the difference between this framework and existing method for MoE?

**Relation To Broader Scientific Literature:**

No

**Theoretical Claims:**

No

---

> ### Author Rebuttal · Authors · 2025-04-01
>
> We thank tR7y for recognizing that "the motivation of the paper is clear" and that "experiments show that the proposed method achieves faster speed and higher performance compared to baseline methods." To address your questions, we provide pointwise responses below.
>
> **[Experiments & Analysis 1: Larger MoE and fewer-expert models]**
>
> Occult is more feasible for fine-grained MoE-LLMs such as DeepSeek-MoE. 16B is a common choice, and larger models usually contain hundreds of billion parameters ( such as DeepSeek-V2 ), which is beyond our capability.
>
> We've conducted additional experiments on Mixtral (8 experts) to demonstrate generalizability to models with fewer experts. Limited to GPU memory, we only tune the last 2 layers with 80 ( bs ) $\times$ 128 ( seq length ) tokens with 4 GPUs ( 2 experts per GPU ). To illustrate the scalability, we also performed training with 8 GPUs on DeepSeek-MoE, showing that the communication savings can scale with device count for expert parallelism ( EP ). We tune all the MoE layers with 32 ( bs ) $\times$ 128 ( seq length ) tokens.
>
> |Model|# Experts|# Devices for EP|Method|Avg Latency Per Step|Speedup|
> |-|-|-|-|-|-|
> |Mixtral-8x7B|8|4|Megablocks|9.64|1.0|
> |Mixtral-8x7B|8|4|Occult, w/o prune|9.08|1.06|
> |Mixtral-8x7B|8|4|Occult, prune within 1 GPU|8.34|1.13|
> |DeepSeek-MoE|64|8|Megablocks|16.56|1.0|
> |DeepSeek-MoE|64|8|Occult, w/o prune|12.10|1.37|
> |DeepSeek-MoE|64|8|Occult, prune within 3 GPUs|10.95|1.51|
> |DeepSeek-MoE|64|8|Occult, prune within 2 GPUs|9.31|1.78|
> |DeepSeek-MoE|64|8|Occult, prune within 1 GPU|8.50|1.95|
>
>
> We also provide the evaluation results for 8-way expert parallelized DeepSeek-MoE with Occult to demonstrate its effectiveness:
>
> |Task|Strategy|No Tune|Prune within 1 GPU|Prune within 2 GPUs|Prune within 3 GPUs|Prune within 4 GPUs|Prune within 5 GPUs|No Prune|
> |-|-|-|-|-|-|-|-|-|
> |MMLU|Router-based|37.95|35.04|40.41|41.34|41.43|41.19|38.66|
> |MMLU|Sim-based|37.95|33.68|39.80|**41.74**|41.40|41.48|38.66|
> ||
> |OpenBookQA|Router-based|32.20|33.8|36.2|37.2|**37.8**|37.2|34.20|
> |OpenBookQA|Sim-based|32.20|33.4|36.4|36.8|**37.8**|37.2|34.20|
> ||
> |MathQA|Router-based|31.19|32.93|35.08|34.97|35.95|**36.08**|33.77|
> |MathQA|Sim-based|31.19|33.17|34.94|35.51|35.24|35.61|33.77|
>
>
> **[Experiments & Analysis 2: Performance improvement explanation]**
>
> The motivation for our proposed pruning methods is to replace some of the original routed experts with carefully assigned alternatives based on certain rules, so that the modified routing choices can be more grouped within some GPUs rather than dispersed, thereby reducing the all-to-all communication overhead. Visualizations in Fig. 8 demonstrate that two-device pruning preserves more essential collaboration patterns found in the original model, while single-device pruning might lose important expert correlations. Our router- and similarity-based pruning approaches are capable of maintaining the most critical expert collaborations, helping the model retain or even enhance its capabilities as a kind of regularization [1, 2]. This targeted preservation of collaboration patterns explains the performance improvements shown in Tab. 3 and Fig. 7.
>
> **[Question: Difference with existing method for MoE]**
>
> Our framework differs from existing MoE libraries in several fundamental ways, making it more communication-efficient:
> - **Novel data structure for token management**: As described in Sec. 4.1, we introduce "Bidirectional Re-Index Matrix ($BRIM$), a novel data structure for unified data management" which efficiently tracks token states across different processing stages. Unlike existing methods that use general-purpose data structures, $BRIM$ is specially designed for MoE workflow with optimized memory access patterns.
> - **State-based token representation**: As stated in lines 206-216, "we outline them as 3 states: Original ($\texttt{ORI}$)... Simplified ($\texttt{SFD}$)... Expanded ($\texttt{EPD}$)," allowing us to maintain tokens in "the token counts across states follow $\texttt{ORI}<\texttt{SFD}<\texttt{EPD}$." This contrasts with existing methods that simply replicate $k$ times for each token, regardless of collaboration patterns.
> - **Two-stage aggregation**: As described in lines 199-203, we implement "summing the intra-collaboration results before all-to-all, and summing the inter-collaboration results after all-to-all, for each token $x$." This two-stage token aggregation enables symmetrical efficient all-to-all communication for both *dispatch* and *combine* operations in MoE pipeline.
>
> Thank you again for your questions. We will emphasize these distinctions in our revised draft.
>
>
>
> [1] Giles, C.L. and Omlin, C.W., 1994. Pruning recurrent neural networks for improved generalization performance. IEEE Transactions on neural networks, 5(5), pp.848-851.
>
> [2] Han, S., Mao, H. and Dally, W.J., 2015. Deep compression: Compressing deep neural networks with pruning, trained quantization and Huffman coding. arXiv preprint arXiv:1510.00149.

---

### Decision · Program_Chairs · 2025-05-01

**Decision:**

Accept (poster)

**Comment:**

All four reviewers converge toward a positive recommendation, with two accepts and two weak accepts. The authors provided comprehensive responses that effectively addressed concerns regarding scalability and experimental clarity. The consensus among the reviewers, along with the paper’s clear motivation in tackling communication costs in MoE models, supports acceptance.